# Long-Short Decision Transformer: Bridging Global and Local Dependencies for Generalized Decision-Making

**Jincheng Wang**[1]    **Penny Karanasou**[2]    **Pengyuan Wei**[1]    **Elia Gatti**[1]
**Diego Martinez Plasencia**[1]    **Dimitrios Kanoulas**[1]
[1]University College London    [2]University of Cambridge
ucabj46@ucl.ac.uk, pk407@cam.ac.uk
{pengyuan.wei.22, elia.gatti, d.plasencia, d.kanoulas}@ucl.ac.uk

## Abstract

Decision Transformers (DTs) effectively capture long-range dependencies using self-attention but struggle with fine-grained local relationships, especially the Markovian properties in many offline-RL datasets. Conversely, Decision ConvFormer (DC) utilizes convolutional filters for capturing local patterns but shows limitations in tasks demanding long-term dependencies, such as Maze2d. To address these limitations and leverage both strengths, we propose the Long-Short Decision Transformer (LSDT), a general-purpose architecture to effectively capture global and local dependencies across two specialized parallel branches (self-attention and convolution). We explore how these branches complement each other by modeling various ranged dependencies across different environments, and compare it against other baselines. Experimental results demonstrate our LSDT achieves state-of-the-art performance and notable gains over the standard DT in D4RL offline RL benchmark. Leveraging the parallel architecture, LSDT performs consistently on diverse datasets, including Markovian and non-Markovian. We also demonstrate the flexibility of LSDT's architecture, where its specialized branches can be replaced or integrated into models like DC to improve performance in capturing diverse dependencies. Finally, we also highlight the role of goal states in improving decision-making for goal-reaching tasks like Antmaze.

## 1 Introduction

Transformers (Vaswani et al., 2017) have emerged as a predominant model, due to their simplicity and capability of modeling high-dimensional distributions (Yu et al., 2022). They serve as the foundational model in various domains, such as the Swin-Transformer (Liu et al., 2021) in Computer Vision (CV) or GPT (Brown et al., 2020) in Natural Language Processing (NLP). In NLP, they significantly enhance the machine's ability to understand and generate natural language by adeptly capturing long-distance dependencies between words. Also, Transformers have been explored as a powerful tool in Reinforcement Learning (RL). The Decision Transformer (DT) (Chen et al., 2021) and its variants (Zheng et al., 2022), conceptualize RL problems as conditional sequence modeling rather than conventional bootstrapping methods (Mao et al., 2022; Liu et al., 2023), drawing parallels with sequence-to-sequence tasks in NLP. DT employs a causal transformer to predict future actions. Leveraging the self-attention mechanism, it can model a wide distribution of behaviors, achieving superior performance in many offline RL tasks (Paster et al., 2022).

Although Transformers have demonstrated impressive performance, its architecture shows limitations in extracting fine-grained local patterns in datasets from other domains(Gulati et al., 2020; Shen et al., 2023), such as the correlations of neighboring pixels in CV (Wu et al., 2021). In the offline RL domain, this architecture struggles to effectively capture inherent Markovian patterns (a local association between two consecutive time-steps) in many RL datasets (Kim et al., 2023). An effective way to address this limitation is to integrate Convolutional Neural Networks (CNNs) into the transformer architecture, as CNNs are good at extracting fine-grained local features (Alzubaidi et al., 2021; Chua & Roska, 1993). Kim et al. (2023) propose Decision ConvFormer (DC), which

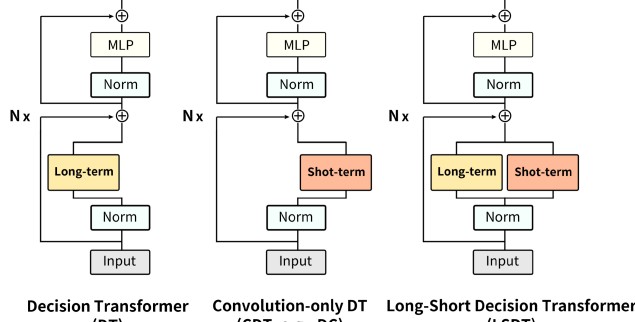

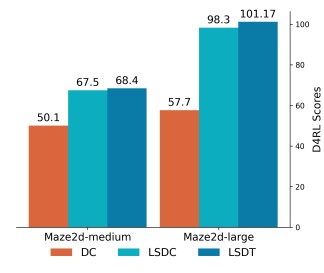

Figure 2: Results in Maze2d. Benefiting from our Long-Short structure, DC gains notable improvement.

Figure 1: The overall architecture of DT,CDT and LSDT.

replaces the attention module with causal convolution filters to extract local Markovian associations. However, DC's architecture has limited adaptability to datasets collected by non-Markovian policies, where long-term dependencies are crucial for optimal decision-making. This limitation is clearly reflected in our results from domains like Maze2d (see Figure 2). These findings highlight the importance of a model's ability to capture both local and global dependencies, which is crucial for enabling agents to learn an optimal policy from diverse datasets. Typically, in offline RL, datasets are often composed of trajectories generated by various policies, leading to data that is not strictly Markovian and may instead exhibit non-Markovian or mixed characteristics. To enhance Transformers' capability in capturing both local and global dependencies, dual branch architectures have been proposed in CV and NLP (Peng et al., 2022; Wu et al., 2020). While these models have proven effective in their respective domains, it is unclear whether such a dual branch structure can also enhance DT's decision-making capabilities in offline RL or help DT and its variants generalize to diverse datasets with different temporal dependencies, such as Markovian and non-Markovian.

In this paper, we propose the Long-Short Decision Transformer (LSDT), a generalized and flexible architecture designed to effectively model both long-term and short-term information. LSDT features two parallel, specialized and customizable branches as shown in Figure 1. To demonstrate the effectiveness of LSDT, we adopted two generic approaches as branches: self-attention and Dynamic Convolution. Specifically, the long-term branch leverages the self-attention mechanism to capture long-range dependencies, whereas the short-term branch employs a parameter-efficient Dynamic Convolution (Wu et al., 2019) to model short-range dependencies. Contrary to fixed dimension allocation to each branch (Peng et al., 2022), our short-term branch can be seen as a dedicated head within the multi-headed attention module (Vaswani et al., 2017). By adjusting the dimension proportions fed into the short-term head, the captured global and local dependencies are varied, enhancing its flexibility to adapt to diverse task demands. The architecture allows for switching to attention-only and convolution-only DT as required by specific scenarios. Moreover, DT struggles in goal-reaching tasks with diverse goals and sparse rewards, where conditioning solely on desired returns is insufficient to guide the agent's behavior. To address this, we introduce goal-state conditioning that explicitly incorporates goal states into the conditioning information, thereby improving the performance of DTs in these tasks.

We conduct evaluations of our proposed approaches on the standard D4RL benchmark (Fu et al., 2020). Results from our studies demonstrate consistent enhancement and superior performance of our LSDT in decision-making compared to DT and its variants. LSDT achieves comparable performance compared to state-of-the-art RL methods. Additionally, the goal-state conditioning method significantly enhances the performance of transformer-based RL, as shown in Antmaze and Maze2d tasks. To validate the flexibility of LSDT, we replaced the Dynamic Conv in the short-term branch with DC's filters, and the observed improvements (e.g., Figure 2) on non-Markovian datasets confirm that our approach is straightforward and plug-and-play with minimal adjustments.

## 2 PRELIMINARIES

**Offline Reinforcement Learning**. In RL, the learning environment can be considered as a Markov Decision Process (MDP), defined by tuples $(S, A, P, R)$. Here, $S$ represents the set of possible states, $A$ represents the set of actions, $P$ represents the state transition probabilities $P(s'|s, a)$, and $R$

represents the reward function $r = R(s, a)$. Unlike online RL, where an agent incrementally learns from interactions with the environment, offline RL involves learning from a static dataset of pre-collected trajectories from diverse behavioral policies $\tau = (s_0, a_0, r_0, s_1, a_1, r_1, \ldots, s_T, a_T, r_T)$. where $s_t, a_t$, and $r_t$ denote the state, action, and reward at time $t$, with $T$ marking the episode's termination. The objective in offline RL is to learn a policy that maximizes expected returns, captured by $\max_\pi E[\sum_{t=1}^T r_t]$, without the permission for further exploration or data gathering. This setting exposes algorithms to challenges such as model generalization and distribution drift, which are inherent in offline reinforcement learning environments. In our approach, we apply a modified transformer architecture in offline RL to learn an optimal policy from suboptimal datasets.

**Decision Transformer**. Our approach is built based on the Decision Transformer (DT) (Chen et al., 2021) architecture, which reframes the RL problem as a conditional sequence modeling task. Unlike other classical RL methods that primarily focus on estimating value functions or computing policy gradients, DT generates an action $a_t$ at time $t$ conditioned on the context of the latest $k$ timesteps, current state, and current return-to-go. The input of DT is formulated as: $\tau = (\hat{R}_{t-k+1}, s_{t-k+1}, a_{t-k+1}, \ldots, \hat{R}_{t-1}, s_{t-1}, a_{t-1}, \hat{R}_t, s_t)$. $\hat{R}_t$ is return-to-go (RTG) at time $t$, computed as $\hat{R}_t = \sum_{t'=t}^T r_{t'}$, where $r$ is rewards, $s$ is states, $a$ is action, $K$ is a hyperparameter and is also referred as the context length. DT parameterized policies by GPT (Radford et al., 2018) which is built based on the decoder of Transformer (Vaswani et al., 2017) architecture. This framework leverages a causal mask and stacked self-attention layers to autoregressively generate action sequences, offering a novel approach in generating policies in RL. A key feature of DT is its exploitation of the self-attention layers, which enables the model to capture long-range dependencies in sequences effectively. This capacity allows DT to directly assign credit to actions, thereby bypassing the slower process of reward propagation through Bellman backups.

**Dynamic Convolutions**. Dynamic Convolution (DynamicConv) (Wu et al., 2019) is a variation of Depthwise Convolution, which performs competitively to the self-attention in transformer architecture. Unlike self-attention, which computes attention weights by comparing each context element across all time steps, DynamicConv predicts separate convolution kernels based on each time step. This module is simpler and more efficient than self-attention, reducing computational complexity from quadratic to linear with respect to input length. For the $t$-th element with output channel $c$, the computation rule is defined as:

$$\text{DynamicConv}(X, t, c) = \sum_{j=1}^k \text{softmax}(f(X_t)_{\frac{cH}{d}, j}) \cdot X_{(t+j-\frac{K+1}{2}), c} \tag{1}$$

where $X$ represents the input to the convolution, $d$ is the input dimension, $k$ is the kernel width, $f$ is a linear module of current timestep only, $j$ is the kernel index indicating which kernel element is being used in the convolution and $H$ is a hyperparameter of weight sharing. DynamicConv is depthwise separable, SoftMax-normalized and share weights across the channel dimension, reducing the number of parameters in the convolutional part from $d^2 \times k$ to $H \times k$ compared to regular convolutions. The configuration endows the model with better generalization and parameter-efficient than standard self-attention. Moreover, exploiting local features dynamically is shown to be beneficial in NLP (Wu et al., 2019). Motivated by these advantages, we explore applying dynamic convolution in offline RL with a transformer, enhancing the model's ability to capture fine-grained local features.

## 3 METHOD

In this section, we focus on two key improvements. First, we propose Long-Short Decision Transformer, designed to effectively capture both global and local dependencies. Second, we introduce the goal-state conditioning method to enhance DTs performance in goal-reaching tasks.

### 3.1 MOTIVATION

To address the limitations of attention-only and convolution-only structures, we propose a long-short structure that leverages the complementary strengths of both, rather than relying on a single type of feature extractor. This long-short structure is designed to effectively handle both Markovian (short-term) and non-Markovian (long-term) dependencies across diverse offline datasets. In order

to intuitively illustrate the differences in captured dependencies between these models, we conducted experiments on the Antmaze-medium dataset (non-Markovian) and the Halfcheetah-medium-expert dataset (Markovian), visualizing the attention scores from the first layer with context length $K$ from the first layer in Figure 3. These scores can be interpreted as alignment measures that indicate how strongly each target token is associated with source tokens (Hu, 2020).

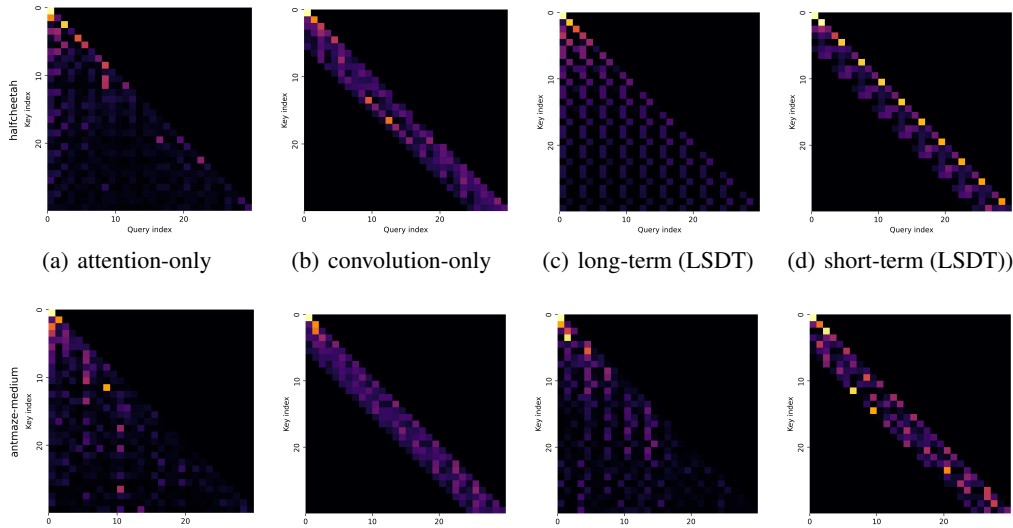

(a) attention-only     (b) convolution-only     (c) long-term (LSDT)     (d) short-term (LSDT))

Figure 3: The heatmap for a single sequence on Antmaze-medium (upper panel) and Halfcheetah-medium-replay (lower panel). Attention scores of (a) attention-only DT, (b) convolution-only DT, (c) the long-term branch of LSDT and (d) the short-term branch of LSDT

The attention matrix demonstrates the captured associations at timestep $t = 30$, where the elements on the horizontal and vertical axes correspond one-to-one with $(\hat{R}_0, s_0, a_0, \ldots, \hat{R}_{29}, s_{29}, a_{29})$. The shape of these matrices is in a lower triangular form because, as a decision-making model, DT can only rely on historical information to generate future action, with all future information masked out (Figure 3(a)). To impose Markov assumption on DT, convolution-only DT (e.g., DC) further narrows the scope of DT's view to adjacent timesteps by setting the convolutional filter size $L = 6$. This operation forms a lower triangular banded pattern as Figure 3(b). Self-attention mechanisms are capable of modeling local associations. In D4RL, datasets can be mainly categorized into two types: Markovian and non-Markovian datasets. Ideally, it should 'pay more attention' to elements between two adjacent timesteps in Markovian offline RL datasets to form a concentrated and banded shape similar to Figure 3(b). Unfortunately, the attention scores in both Antmaze and Halfcheetah are dispersed, indicating that DT tends to prioritize long-range dependencies over local patterns. In convolution-only DT, the available historical information is restricted, which prevents the model from effectively capturing global dependencies in non-Markovian datasets. While a larger kernel size can be used to expand the receptive field of models, CNNs generally struggle to model long-term dependencies (Gulati et al., 2020) leading to worse performance, as we discussed in Appendix 7. The aim of our LSDT is to leverage the unique strengths of both branches, allowing them to complement each other and thereby enhance the model's generalization across diverse datasets. Specifically, the long-term branch is designed to capture global dependencies by identifying which distant information in the sequence helps predict the next action, while the short-term branch focuses more on local features, including but not limited to Markov patterns. For instance, in Halfcheetah, our short-term branch captured the Markov property (Figure 3(d) upper part), where excluding the element's self-correlation, the next state $s_{t+1}$ only depends on the previous state $s_t$. Conversely, in the Antmaze environment (Figure 3(d) lower part), it captures more complex local patterns.

## 3.2 LONG-SHORT DECISION TRANSFORMER

Inspired by the developments of dual-branch structure in language (Zhu et al., 2021) and vision tasks (Peng et al., 2022), our approach seeks to employ a similar strategy to enhance the capability of exploiting local fine-grained patterns of GPT architecture used in DT. To this end, we present

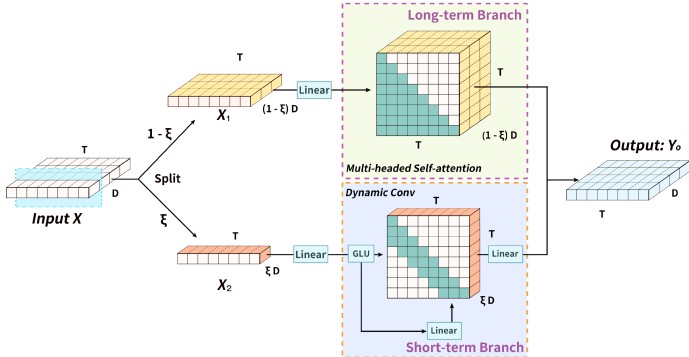

Figure 4: The architecture of Masked Long-Short block.

Long-Short Decision Transformer (LSDT). The key innovation in LSDT lies in replacing the standard masked attention block with our Masked Hybrid block, which features a simple dual-branch structure as shown in Figure 1. By integrating an additional convolution branch into DT, our approach allows the model to better capture local features, while effectively combining it with the original attention branch's strength in modeling global dependencies. This comprehensive ability is particularly crucial in RL tasks, where the ability to discern and respond to immediate information while maintaining the view of the overall information is essential for decision-making (Chen et al., 2023; Bakker, 2001). Furthermore, LSDT is a general architecture, it can be transformed into original attention-only DT or convolution-only DT by adjusting the input dimension to each branch. We keep most settings as DT, including training and inference stages, except for the inner architecture.

### 3.2.1 MASKED LONG-SHORT BLOCK

Our detailed Masked Long-Short block consists of two parallel branches, shown in Figure 4. This structure leverages the complementary strengths of its two branches. Contrasting with methods that either feed the whole input to each branch or evenly split the input for the two branches (Peng et al., 2022; Wu et al., 2020), our approach splits the input $X$ along the channel dimension asymmetrically. Consider a input vector $X$ with hidden dimension $D$ and sequence length $T$, we partition the dimension $D$ into two parts according to the dimension ratio $1 - \xi$ (for long-term branch) and $\xi$ (for short-term branch), denoted as $X_1$ and $X_2$. This division allows us to flexibly adjust the dimensions according to the specific requirements of different tasks, aiming to achieve better performance. We empirically demonstrate the impact of different dimension ratios $\xi$ in Section 4.5. The $X_1$ is fed into the attention branch to focus on capturing complex hidden relationships across sequences, while the other part $X_2$ is processed by a convolution branch along the diagonal to capture local feature patterns. For the short-term branch, we use DynamicConv, detailed in Section 2. DynamicConv is characterized by fewer parameters and enhanced performance, dynamically generating weights based on current input instead of using fixed weights after training. These strengths endow the model with greater flexibility and generalization capabilities, beneficial for decision-making in RL.

### 3.3 GOAL-STATE CONDITIONING

DT generates optimal paths from sub-optimal trajectories by conditioning on the desired sum of the future reward (*return-to-go, RTG*) and contextual information, utilizing original data representations from the D4RL dataset (Chen et al., 2021; Wu et al., 2023). However, DT faces significant path planning challenges in goal-reaching tasks with diverse goals and sparse rewards, where based solely on returns as goals are inadequate for directing agents on where to go or which subpart to stitch. An example of DT failing to generate a path to the target location, even from the expert training dataset, is shown in Figure 5. Consider a path planning task to a target position **x** with a reward of 1 at **x** and 0 otherwise. The training data includes the optimal paths to goal **x**, goal **y**, and the trajectories that pass through goal **y** on the way to goal **x**. In DT setup, the RTG for all successful training trajectories are marked as 1. However, such processing can misguide the agent in environments with diverse goals, leading to planning failures. The three potential paths generated by DT, conditioning solely on an RTG of 1 leading to goal **x**, are depicted as 'Generation A' with dotted lines. Moreover, previous works (Emmons et al., 2021) categorize RL algorithms by different types of conditioning

information. In some experiments, DT is treated as a reward-conditioned algorithm and directly compared with other goal-conditioned methods, which may lead to an unfair comparison.

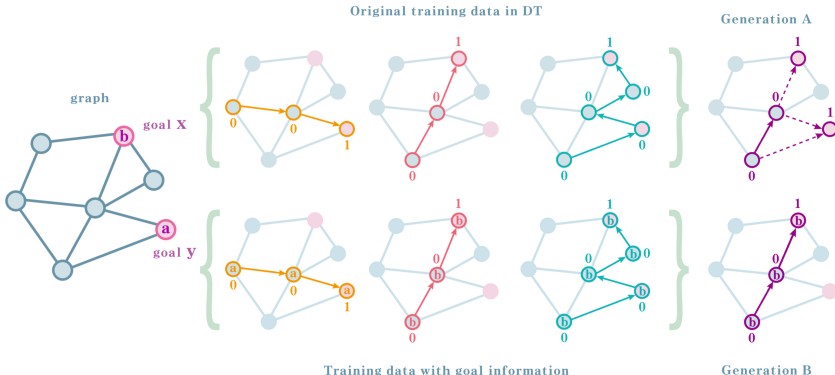

Figure 5: An example demonstrating the impact of conditioning information on path planning in environments with diverse goals and sparse rewards. The top panel shows how an agent struggles without target information, leading to a misguided goal. The bottom panel depicts the agent correctly finding its direction when trained on a dataset where the goal position is incorporated into the state.

To address these, we introduce this 'Goal-State Conditioning' method which transforms goal-conditioned DT into a model conditioned on both the goal and the goal state. We also demonstrate why relying solely on original reward conditioning is insufficient for handling complex goal-reaching tasks, such as Antmaze-medium in Appendix B.1. Although this simple idea has been explored in several goal-conditioned MLP-based models (Liu et al., 2022; Paster et al., 2022; Schaul et al., 2015), it is less researched and discussed in sequence modeling RL. We also extend this approach to other transformer-based RL methods (Section 4.3). This integration provides a more comprehensive basis to model for path planning. The successful path leading to goal **x**, generated using this approach, is depicted as 'Generation B' in Figure 5.

### 3.3.1 IMPLEMENTATION OF GOAL-STATE CONDITIONING

During the training stage, we augment each state in the dataset with the corresponding goal state from its trajectory (existing in D4RL dataset), indicating to the agent which segments can be stitched together to generate an optimal path for each distinct goal. We next formally detail the modifications made to the state representation. Let $s_t \in \mathbb{R}^n$ denote the original state vector at time step $t$, where $n$ represents the dimensionality of the state vector. Let $g \in \mathbb{R}^m$ be the goal state's coordinate vector, with $m$ being the dimensionality of the goal state coordinates. Through Goal-State Concatenation, we obtain an augmented state vector $S'_t$ at time step $t$, which can be defined as: $S'_t = [s_t, g]^T$, where the augmented state vector $S'_t \in \mathbb{R}^{n+m}$ is formed by stacking the original state vector $s_t$ and the goal state vector $g$. Furthermore, with the incorporation of goal state information, the representation of modified trajectory $T_g$ in offline training dataset is represented as: $T_g = (r_0, S'_0, a_0, \ldots, r_T, S'_T, a_T)$, where $T$ is the length of trajectory, $a$ is action and $r$ is reward.

During the evaluation, our process mainly adheres to DT with a key modification to enrich condition information. Initially, we specify the desired performance and the starting state, then concatenate the target goal coordinates with the state to create an augmented state representation. This augmented state, along with the desired performance, serves as the conditioning information to initiate trajectory generation. Subsequently, each new observed state $S_t$ is concatenated with the goal state $g$ to form the augmented state $S'_t$ and insert them into history. We then repeatedly feed the last $K$ historical timesteps into the model to autoregressively generate the next action until episode termination.

## 4 EXPERIMENTS

Our experiments systematically assessed the proposed LSDT and goal-state conditioning approaches around the following questions:

Table 1: MuJoCo Evaluation Results. For locomotion tasks, the mean values of 5 random seeds are reported, with 20 evaluations each. For Antmaze, values are averaged over 4 seeds with 100 evaluations each, using goal-state conditioning. The values within 5% of the best scores per task are highlighted in bold, in line with IQL (Kostrikov et al., 2021) paper. '-m' stands for *medium*, '-m-r' stands for *medium-replay*, and '-m-e' stands for *medium-expert* dataset. '-d' denotes *diverse*.

| Dataset | TD3+BC | IQL | CQL | RvS | DT | QDT | DC | LSDT |
|---|---|---|---|---|---|---|---|---|
| Halfcheetah-m | **48.3** | **47.4** | 44.0 | 41.6 | 42.6 | 42.3 | 43.0 | 43.6 |
| Hopper-m | 59.3 | 66.3 | 58.5 | 60.2 | 67.6 | 57.2 | **92.6** | 87.2 |
| Walker2d-m | **83.7** | 78.3 | 72.5 | 71.7 | 74.0 | 67.5 | 79.2 | **81.0** |
| Halfcheetah-m-r | **44.6** | **44.2** | 37.5 | 38.0 | 36.6 | 30.0 | 41.3 | **42.9** |
| Hopper-m-r | 60.9 | **94.7** | 95.0 | 73.5 | 82.7 | 45.8 | **94.2** | **93.9** |
| Walker2d-m-r | 81.8 | **73.9** | 77.2 | 60.6 | 66.6 | 30.3 | 76.6 | **74.7** |
| Halfcheetah-m-e | **90.7** | 86.7 | 91.6 | **92.2** | 86.8 | - | **93.0** | **93.2** |
| Hopper-m-e | 98 | 91.5 | 105.4 | 101.7 | **107.6** | - | 110.4 | **111.7** |
| Walker2d-m-e | **110.1** | **109.6** | 108.8 | 106.0 | 108.1 | - | 109.6 | **109.8** |
| Locomotion average | 75.3 | 76.5 | 77.6 | 71.7 | 76.4 | - | **82.2** | **82.0** |
| Antmaze-umaze | 78.6 | **87.5** | 74.0 | 65.4 | 69.8 | - | **85.0** | 80.0 |
| Antmaze-umaze-d | 71.4 | 62.2 | **84.0** | 60.9 | 70.3 | - | 78.5 | **83.2** |
| Antmaze-m-play | 10.6 | 71.2 | 61.2 | 58.1 | 0.0 | - | 1.5 | **85.5** |
| Antmaze-m-d | 3.0 | 70.0 | 53.7 | 67.3 | 0.0 | - | 0.0 | **75.8** |

- How does LSDT perform against DT and its variants? (4.2,4.3)

- How does the different channel dimension ratio $\xi$ affect the performance? (4.5)

- Is goal-state conditioning effective for enhancing the path planning capability of DTs? (4.3)

- Can other algorithms leverage the advantage of our Long-Short architecture? (4.4)

We evaluate our approach using continuous control tasks in OpenAI Gym, serving as a benchmark for offline RL algorithm performance. Through these experiments, we aim to quantify our method's enhancements in diverse datasets. Details of benchmarks and datasets can be found in Appendix A.

## 4.1 OFFLINE RL BASELINES

We compare our approach with representative methods in value-based offline RL, including TD3+BC (Beeson & Montana, 2022), IQL (Kostrikov et al., 2021), and CQL (Kumar et al., 2020). Additionally, to demonstrate the enhancement offered by our method, we compare LSDT with three DT-based methods: the standard DT (Chen et al., 2021), Q-learning Decision Transformer (QDT) (Yamagata et al., 2023), and Decision ConvFormer (DC) (Wu et al., 2019). Similar to our convolution-only DT, DC achieves superior performance by capturing local features with convolution filters. Moreover, we include comparisons with the goal-conditioned RvS (Paster et al., 2022) to illustrate the role of goal-state conditioning in DT. In Maze2d, we primarily compare with QDT, which significantly outperforms DT.

## 4.2 LOCOMOTION TASKS WITH MARKOVIAN DATASETS

We evaluate LSDT in Hopper, Walker2d and HalfCheetah with dense rewards. The results of our experiments are presented in Table 1. All reported scores are normalized following the instructions in D4RL (Fu et al., 2020). The QDT values are reported from (Yamagata et al., 2023), DC values from (Wu et al., 2019), and other values from IQL (Kostrikov et al., 2021) paper. Our LSDT consistently achieves or approaches state-of-the-art performance in all tasks, demonstrating the effectiveness of our Long-Short architecture. Notably, LSDT is the only approach that consistently shows advanced performance across all *medium-replay* and *medium-expert* datasets. Although LSDT does not surpass value-based algorithms such as TD3+BC and IQL on 'Halfcheetah-medium', it achieves the highest scores among DT and its variants. Finally, the overall performance of both DC and our

Table 2: Maze2d Evaluation Results. We report the mean and standard deviation of 4 random seeds, with 100 evaluations each. CQL, DT and QDT values are reported from QDT paper (Yamagata et al., 2023). The best values are highlighted in bold.'-u','-m' and '-l' denote *umaze*, *medium*, and *large*, respectively.

| Dataset | CQL | DT | QDT | $DC_h$ | $DC_h$-gs | LSDT | LSDT-gs |
|---|---|---|---|---|---|---|---|
| Maze2d-u | **94.7** | 31.0 | $82.9 \pm 8.8$ | $10.9 \pm 2.2$ | $68.6 \pm 1.4$ | $9.6 \pm 2.6$ | $72.3 \pm 1.1$ |
| Maze2d-m | 41.8 | 8.2 | $48.5 \pm 9.4$ | $16.3 \pm 0.5$ | $50.1 \pm 4.5$ | $17.2 \pm 2.3$ | **$68.4 \pm 3.9$** |
| Maze2d-l | 49.6 | 2.3 | $62.0 \pm 13.4$ | $20.2 \pm 4.6$ | $57.7 \pm 12.3$ | $12.4 \pm 3.2$ | **$116.7 \pm 29.7$** |

LSDT suggests that including specialized modules to focus on short-term information can significantly enhance DT's performance on Markovian datasets.

### 4.3 MAZE2D AND ANTMAZE TASKS WITH NON-MARKOVIAN DATASETS

To evaluate the stitching and long-term credit assignment capabilities of LSDT on non-Markovian datasets, as well as to demonstrate the effectiveness of 'goal-state conditioning' method in environments characterized by diverse goals and sparse rewards, we assess our model using the Maze2d and more complex AntMaze environments. To ensure a fair comparison, we conduct experiments on hybrid DC ($DC_h$) presented in DC (Kim et al., 2023). It replaces the convolution block with an attention block in the final layer, enhancing its ability to manage long-term credit assignment. In tables, '-gs' stands for goal-state conditioning.

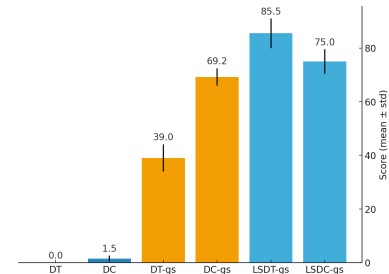

Figure 6: Ablation study investigating the effects of goal-state concatenation and the long-short structure in DT and DC on Antmaze-medium-replay.

Both results from ablation studies in Table 2 and Figure 6 demonstrate that simply offering models with additional conditioning information - goal's position can dramatically improve the performance of DT and its variants. By doing so, the agent gains a better understanding of the trajectory towards the desired location. This finding validates the effectiveness and generalization of goal-state conditioning in transformer-based models. Moreover, we further conduct an empirical experiment to show how goal-state conditioning influences the generated path in Appendix B.1.

Across all results in Table 1 and Table 2, LSDT achieves superior performance in environments with non-Markovian datasets. For Maze2d, CQL, and QDT perform better in the *umaze* environment, benefiting from Q-learning's efficient backward reward propagation in tasks with shorter trajectories. However, in the more complex *Medium* and *Large* environments, our LSDT outperforms other approaches. In Antmaze tasks, the scores highlight the effectiveness and generalization of our long-short structure in complex tasks, particularly in the *Medium* tasks where DT is minimally effective. The comparison of DT-gs and LSDT-gs in Figure 6 illustrates that integrating an additional local feature extractor into DT can assist the model in making better decisions.

### 4.4 ENHANCING OTHER ALGORITHMS BY COMBINING OUR ARCHITECTURE

In this section, we explore whether our dual-branch structure can be integrated into other approaches to enhance their performance. We present the Long-Short Decision ConvFormer (LSDC) as a demonstration, which substitutes the Dynamic-Conv in the short-term branch with DC's convolution filters. Given that $DC_h$ aims to enhance long-term credit assignment over DC, we aim to assess whether our architecture outperforms the alternative of replacing the last block with an attention block for capturing global dependencies.

| Dataset | $DC_h$ | LSDC | LSDT |
|---|---|---|---|
| M2d-u | 68.6 | 67.7 | **72.3** |
| M2d-m | 50.1 | **67.5** | **68.4** |
| M2d-l | 57.7 | 98.3 | **101.17** |

Table 3: Comparisons of $DC_h$, LSDT and LSDC in Maze2d domain.

We conduct experiments on Maze2d and Antmaze across 4 seeds. Table 3 and Figure 6 show that LSDC achieves competitive performance, and significantly surpasses the original DC method in non-Markovian datasets. In Maze2d, larger mazes show more significant performance gains, highlighting our approach's potential to enhance other models by integrating a dual branch structure that effectively captures both global and local dependencies.

## 4.5 ABLATION STUDY ON DIMENSION RATIO $\xi$

In our LSDT, we introduce the channel dimension ratio $\xi$ rather than evenly splitting the dimension, aiming to allow the model to adapt better to different tasks and achieve improved performance. Different channel dimension ratios will lead the model to focus on different features, thereby affecting dependency modeling. To assess the importance of this setting, we evaluate LSDT with varying dimension ratios ranging from (0%-100%). When the convolution dimension ratio is set to 0%, the model is attention-only, reverting to standard DT. When the ratio reaches 100%, all input dimensions are processed by the short-term branch, transforming the model into convolution-only DT (CDT).

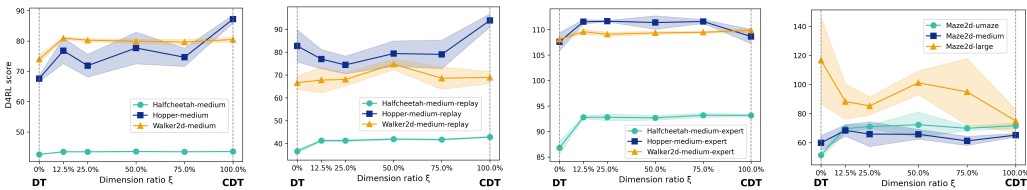

Figure 7: Performance of the locomotion environments across different dimension ratios $\xi$. The plots display mean values and standard deviations over 5 random seeds, each with 20 rollouts. From left to right: *medium* dataset, *medium-replay*, *medium-expert*, *maze2d.*

Figure 7 shows how the ratio influences the final performance across different environments and datasets. Except for the *Hopper-medium-replay* dataset and *Maze2d-Large*, the performance of the model is enhanced by integrating a local feature extractor into DT across all dimension ratios. On HalfCheetah, there is no significant further improvement in performance beyond a convolution dimension ratio of 12.5% on every dataset. For Walker2d, LSDT achieves its best performance on each dataset at convolution dimension ratios of 12.5%, 50%, and 12.5% respectively. The observations from the Hopper task are interesting, LSDT reaches its highest scores at a ratio of 100% on *medium* and *medium-replay*, suggesting that the global patterns from the attention branch are totally harmful to decision-making. However, on *medium-expert* dataset, relying solely on the convolution branch leads to decreased performance. This finding aligns with the performance gap observed between DC and our LSDT on Hopper tasks in Table 1. DC is specifically designed to mainly focus on local patterns, which causes our LSDT to lag slightly behind on the *medium* and *medium-replay* datasets. However, on *medium-expert*, LSDT achieves slightly higher performance by integrating long-term information. This suggests that LSDT performs better on datasets with higher-quality data.

Based on ablation studies conducted on Locomotion (Markovian) and Maze2d (non-Markovian) datasets, we propose general guidelines for selecting the dimension ratio based on the dataset type. Additionally, we explore a potential alternative to eliminate the need for manual tuning of the dimension ratio, as discussed in Appendix B.10, to inspire further enhancements to our approach.

- Avoid setting the dimension ratio to 0% when the training datasets are collected by Markovian policies, as nearly all tasks benefit from integrating the additional short-term branch. For such tasks, we recommend choosing a dimension ratio of 50% or higher (e.g., 75%).

- Avoid setting the dimension ratio to 100% when the training datasets are collected by non-Markovian policies, as this limits the model's ability to access long-term historical information necessary for accurately predicting the current token. For such tasks, it is advisable to initially conduct experiments with below (e.g., 0%, 12.5%) or equal to 50%.

- When computational resources are limited, we recommend 50% as the default, balancing performance and tuning cost, especially without prior knowledge about the dataset type.

## 5 RELATED WORK

**Transformers for RL.** Approaches such as Decision Transformer (DT) (Chen et al., 2021) and Trajectory Transformer (Janner et al., 2021) that abstract the offline RL as a sequence modeling problem have been popular (Wu et al., 2023). Compared to the traditional offline RL methods, these approaches are simpler and more robust through doing RL via Supervised learning (RvS) (Emmons et al., 2021) rather than learning a Q-function or computing policy gradients (Kumar et al., 2020; Kostrikov et al., 2021; Beeson & Montana, 2022). DT is among the most successful methods in this field. It generates a sequence of actions through a transformer decoder conditioned on desired outcomes and previous context. Recently, numerous of DT's variants have been developed leveraging the standard architecture. For instance, the Q-learning DT (QDT) (Yamagata et al., 2023), integrates Dynamic Programming with the DT framework to enhance the optimal path generation ability of DT. Another notable development is Decision ConvFormer (DC) (Kim et al., 2023), which replaces attention blocks with convolution filters to more efficiently capture local associations. In contrast to DC, we aim to design a general model structure, focusing on enhancing performance across various tasks by balancing local and global dependencies, rather than limiting attention to local information.

Gated Transformer-XL is a pioneering approach in applying the transformer model to RL, particularly highlighting the importance of architectural modifications to address optimization challenges inherent in the standard transformer framework (Parisotto et al., 2020). This issue has sparked a wave of research focused on designing transformer-based backbones specifically tailored for RL applications (Mao et al., 2022; Liu et al., 2023; Hu et al., 2023) . In this vein, our work contributes to this field by enhancing the model structure of DT with a dual-branch architecture adept at extracting information within different ranges, making it more effective for RL.

**Convolution strategies in Transformers.** Convolutional Neural Networks (CNNs), known for their efficacy in capturing local feature patterns in vision domain (Kaiser et al., 2018; Yang et al., 2021; Roy et al., 2018). While the transformer is good at modeling global dependencies, it faces challenges in capturing local feature patterns (Jiang et al., 2020; Wu et al., 2021; Su et al., 2022). Therefore, many approaches have been studied to combine the strengths of transformers and CNNs, notably in Conformer and Branchformer (Gulati et al., 2020; Peng et al., 2022). These models have demonstrated improved performance in local feature capture by their success in various tasks across CV and NLP. Similar to the Lite Transformer (Wu et al., 2020), our model employs two branches to model global and local dependencies separately, one branch for standard self-attention and another for convolution. Lite Transformer uses Light Convolution to reduce the computation and model size for mobile applications. Differently, our LSDT is specifically designed to address challenges in offline RL tasks. We utilize Dynamic Convolution (Wu et al., 2019), which offers enhanced generalization capability in capturing local features across diverse RL scenarios by dynamically generating the weights at each timestep instead of fixing them after training.

## 6 CONCLUSION

In this paper, we addressed the challenge of effectively handling both Markovian and non-Markovian properties in offline RL datasets by introducing the Long-Short Decision Transformer (LSDT). It is a general-purpose architecture to effectively capture global and local dependencies across two specialized and customized parallel branches. To demonstrate the effectiveness of this architecture, we adopted two generic approaches (self-attention and Dynamic Convolution) as the long-term branch and the short-term branch. Our experiments on the D4RL benchmarks demonstrate that LSDT consistently achieves strong performance across both Markovian and non-Markovian datasets, validating the effectiveness of the proposed parallel architecture. We show that the dimension ratio of LSDT can affect the dependencies captured by the model and performance. Choosing an appropriate dimension ratio can further enhance decision-making. Furthermore, the performance improvement of DC with the integration of the long-short structure demonstrates the transferability and adaptability of our architecture. The results of applying goal-state conditioning to DT and its variants show that simply incorporating explicit goal information can dramatically improve performance in goal-reaching tasks like Antmaze. All these results demonstrate our architecture is a promising approach, providing possibilities to design more powerful and parameter-efficient LSDT variants by replacing the current branches with more expressive feature extractors. In future work, we aim to extend our methods to more intricate tasks and explore LSDT's potential in multi-task trajectory datasets.

# 7 ACKNOWLEDGMENT

This work was supported by the EPSRC through their Prosperity Partnership program (EP/V037846/1) and by the UKRI Future Leaders Fellowship [MR/V025333/1] (RoboHike). For the purpose of Open Access, the authors have applied a CC BY public copyright license to any Author Accepted Manuscript version arising from this submission.

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

## A  BENCHMARK AND DATASET DETAILS

**OpenAI Gym locomotion.** We evaluate LSDT on OpenAI Gym continuous locomotion tasks (Hopper, Walker2d, HalfCheetah) with dense rewards. Our approach was trained and tested on three 'v2' datasets from D4RL  (Fu et al., 2020) with different quality levels: *medium*, *medium-replay* and *medium-expert* datasets. The *medium* dataset is collected from a policy which is a third of the performance of an expert policy. The *medium-replay*, sourced from this policy's replay buffer. For *medium-expert* dataset, it consists of half of the samples (1 million) from the medium policy and the other half (1 million) from the expert policy. Normally, these tasks are used to evaluate the model's performance across different sets of trajectories of varying quality.

**Maze2d.** Maze2d is a navigation task, requiring a 2D ball to reach a target goal location within a maze. We conduct experiments across three different sizes of environments with sparse rewards: umaze, medium and large, each presenting increasing levels of difficulty. The agent receives a reward of 1 if it successfully reaches the target location and 0 otherwise. The training dataset comprises trajectories of the agent navigating towards random goal locations.

**Antmaze.** This domain utilizes the same maze layout and objective as Maze2d but replaces the 2D ball with a more complex Ant robot. In D4RL, three types of datasets are included for AntMaze: '-umaze' features a fixed start to a specific goal, '-diverse' involves random starts to random goals, and '-play' comprises trajectories originating from a set of hand-selected starting points, leading to another set of hand-picked target locations that may differ from the goal locations used during evaluation. Especially, all of the datasets contain very few near-optimal trajectories, requiring relatively high stitching suboptimal paths capability to generate a path to desired goals.

**Adroit.** The Adroit domain consists of high-dimensional robotic manipulation tasks using a 24-DoF Shadow Hand for actions like hammering, door opening, and pen twirling. It is designed to evaluate fully offline RL under narrow data distributions and sparse rewards. Three types of datasets are used: human demonstrations, expert data from a fine-tuned RL policy, and a 'cloned' dataset combining imitation and demonstration data. Unlike standard Gym MuJoCo tasks, Adroit data involves human demonstrations, sparse rewards, and complex exploration challenges, making it difficult to solve with traditional RL methods. The results in Adroit for LSDT are shown in Appendix C.

## B  FURTHER DISCUSSION

### B.1  HOW DOES GOAL-STATE CONDITIONING AFFECT PATH PLANNING IN ENVIRONMENTS WITH DIVERSE GOALS AND SPARES REWARDS?

To illustrate the challenges of path planning that DT encounters in environments characterized by diverse goals and sparse rewards, we consider the AntMaze medium as a representative example. In addition to intuitively demonstrating how goal-state conditioning effectively leads the agent to target, we also empirically show how insufficient conditioning information misleads agents toward incorrect goals, as theoretically analysis in Figure 5.

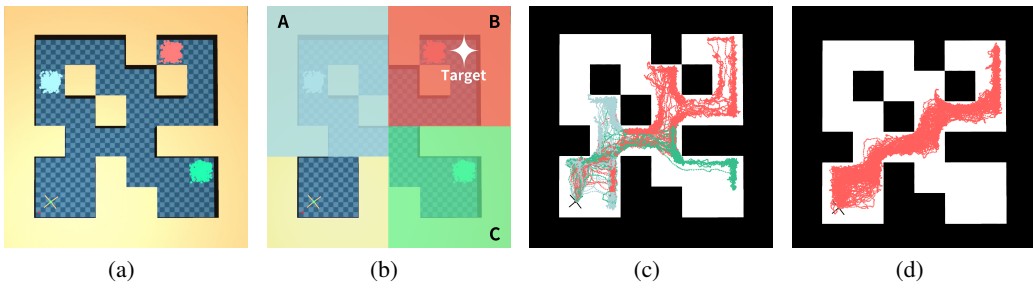

|          |          |          |          |
| :------: | :------: | :------: | :------: |
| (a)      | (b)      | (c)      | (d)      |

Figure 8: Visualization of AntMaze-Medium Task: (a) Distribution of target locations; (b) Zonal divisions for goal-oriented analysis; (c) Trajectory paths generated using DT; (d) Enhanced trajectory visualization with goal-state concatenation.

First, we mark the goal location of all the trajectories from the D4RL dataset in the environment (Figure 8(a)). Second, according to these goals distribution, we divide the environment into three zones as shown in Figure 8(b). For *Medium* environment, the evaluation target location is marked with a white star in Zone **B**. Through this segmentation, the trajectories are classified into three groups, with each path colored according to the zone in which the Ant's final location resides.

We illustrate the trajectories generated by a fully trained DT when conditioned on a desired RTG of 1 in Figure 8(c). It becomes evident that the agent might be incorrectly guided to areas **A** or **C**, which are significantly distant from the intended target area. This misdirection occurs because all RTGs within successful trajectories in the dataset are uniformly labeled as 1, regardless of the actual target location. Therefore, the agent struggles to discern the correct direction during evaluation when conditioned solely on the RTG. Consequently, the termination points of the paths generated by DT are highly dependent on the target locations seen during the training phase. We also show the trajectories generated by DT with our goal-state concatenation approach in Figure 8(d). By integrating the location vector of goals into states, DT becomes capable of correctly planning paths that lead the Ant robot to the target location, which underscores the significance of including the goal vector as additional condition information for effective path planning. We quantify the improvement of DT provided from goal-state conditioning in Table 4.

Table 4: Evaluation results of DT with goal-state conditioning. We report the mean and standard deviation of 4 random seeds, with 100 evaluations each. '-gs' denotes goal-state conditioning.

| Dataset | DT | DT -gs |
|---|---|---|
| antmaze-umaze-v2 | $69.8 \pm 0.2$ | **$77.0 \pm 1.2$** |
| antmaze-umaze-diverse-v2 | $70.3 \pm 5.3$ | **$76.3 \pm 1.5$** |
| antmaze-medium-play-v2 | 0.0 | **$39 \pm 5.1$** |
| antmaze-medium-diverse-v2 | 0.0 | **$46.8 \pm 8.1$** |

## B.2    HOW DOES THE CHANNEL DIMENSION RATIO AFFECT DEPENDENCY MODELING?

To intuitively demonstrate the impact of the dimension ratio $\xi$ on dependency modeling within each branch, we conducted experiments on the Halfcheetah-medium-expert environment, using a hidden dimension of 128, kernel size of 11 and context length of 10. We experiment the dimension ratio for the convolution branch at $75\%$ , $50\%$ , $25\%$ , and $12.5\%$ respectively. Figure 9 shows the learned attention matrix: the convolution branch's matrix is shown in the top panel, and the corresponding matrix for the self-attention branch is in the bottom panel. In our model, each timestep consists of a tuple: return-to-go $\hat{R}_t$, state $s$ and action $a$. The indices for the $x$ and $y$ coordinates are derived from multiple timesteps, represented as sequences $(\hat{R}_0, s_0, a_0, \ldots, \hat{R}_{29}, s_{29}, a_{29})$.

The dimension ratio plays a crucial role in determining the types of patterns each branch captures. For instance, with a convolution dimension ratio of $75\%$, the short-term branch finds that each state is related only to itself and its preceding state. Conversely, the attention branch models dependency across the sequence of actions. At an attention dimension ratio of $75\%$, it identifies three dependencies: the current state depends only on previous states, the current return-to-go depends on previous states, and the current actions are closely linked to the return-to-go. The dependency of actions aligns with the concept of DT, which generate sequences of actions based on desired performance outcomes. These findings demonstrate that varying the dimension ratio results in different dependencies modeled in LSDT. We believe that choosing an appropriate dimension ratio can better tailor the model to the specific requirements of each task, thereby achieving better performance as demonstrated in the experiment section.

## B.3    HYBRID FUSION MODULE

To dynamically select valuable features and effectively integrate information from the two branches, we designed a hybrid fusion module incorporating a competitive selection process. Since this module introduces additional parameters without offering significant improvements over direct concatenation in most tasks and page constraints, we present it as an optional component here to provide interested readers with a potential direction for further exploration. This module explicitly deter-

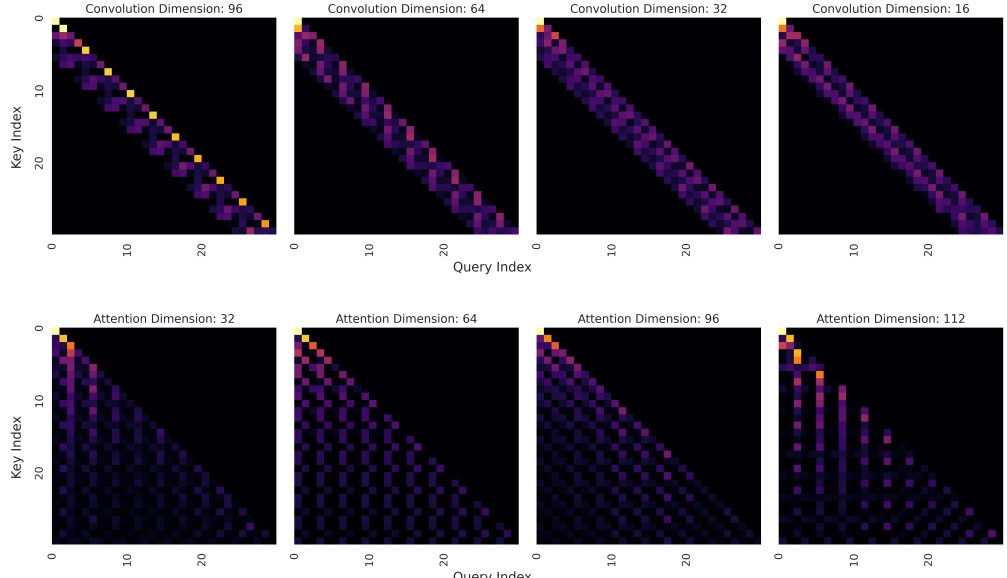

Figure 9: Attention scores of Halfcheetah-medium-expert Task. **Top**: Attention scores of short-term branch (convolution). **Bottom**: Attention scores of short-term branch (self-attention).

mines the relative importance of different channels through a softmax operation, allowing it to highlight the most relevant sub-features in the final output. Importantly, by visualizing these importance weights, we can determine which channels and branches contribute the most critical information for decision-making, providing a clearer insight into the contribution of each branch.

### B.3.1 DETAILS OF HYBRID FUSION MODULE STRUCTURE

Many existing methods (Peng et al., 2022; Wu et al., 2020) primarily employ direct concatenation to integrate features from different branches. However, the importance of features captured by each branch in each timestep is ignored. Inspired by Interformer and SE block (Lai et al., 2023; Hu et al., 2018), we introduce the hybrid fusion module (HFM) to select valuable characteristics dynamically and efficiently fuse local and global representations from two branches. Initially, the outputs from the attention and convolution branches denoted as $Y_{atte} \in R^{T \times (1-\xi)D}$ and $Y_{conv} \in R^{T \times \xi D}$ are concatenated, where $\xi$ is the dimension ratio for short-term branch. The concatenated features $Y$ are then processed through a multi-layer perceptron (MLP) with a ReLU activation, reducing the channel dimension from $D$ to $\frac{D}{2}$ for refining the feature representation:

$$Y = Concat(Y_{atte}, Y_{conv}), \quad \hat{Y} = ReLU(W_f(Y)) \tag{2}$$

where $W_f \in R^{D \times \frac{D}{2}}$ denotes the MLP layer. Next, we restore the refined feature $\hat{Y}$ through two MLP layers, producing two features with the same size as the long-term branch and short-term branch, and then concatenate them for information aggregation.

$$Z = Concat(W_{f1}(\hat{Y}), W_{f2}(\hat{Y})) \tag{3}$$

where $W_{f1} \in R^{\frac{D}{2} \times (1-\xi)D}, W_{f2} \in R^{\frac{D}{2} \times \xi D}$. Finally, we employ the softmax to weight the aggregation feature $Z$ in time dimension, learning a selective weight matrix $A \in R^{T \times D}$. By multiplying the matrix $A$ with original output $Y$ point-wisely, we can focus on important features and filter out useless information from $Y$ dynamically in each timestep. The output $Y_o$ is generated as follow:

$$Y_o = Softmax(Z) \cdot Y \tag{4}$$

### B.3.2 COMPARISON OF HFM AND DIRECT CONCATENATION

To assess the effectiveness of our proposed hybrid fusion module (HFM), we compare HFM with direct concatenation fusion method. We perform experiments on four different tasks using datasets of varying quality: Walker2d-medium, Hopper-medium-expert, Halfcheetah-medium-replay, Maze2d-medium. Figure 10 (left) and Figure 10 (right) compare the learning curves and performance of our HFM with direct concatenation, respectively. We can observe that HFM can stabilize the training process and enhance performance.

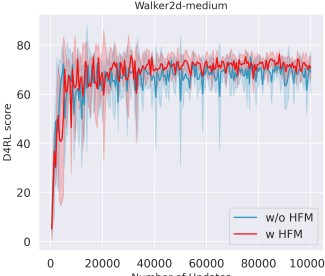

|  | w/o | w |
|---|---|---|
| Walker2d-m | 79.2±1.8 | **80.3±0.6** |
| Halfcheetah-m-r | 41.1±0.2 | **41.9±0.3** |
| Hopper-m-e | 111.5± 0.4 | **111.7±0.1** |
| Maze2d-m | 64.7± 5.6 | **68.4± 3.9** |

Figure 10: Ablation on HFM. **Left:** comparison of training curves on Walker2d-medium. **Right:** performance comparison. Experiments conducted across 4 seeds.

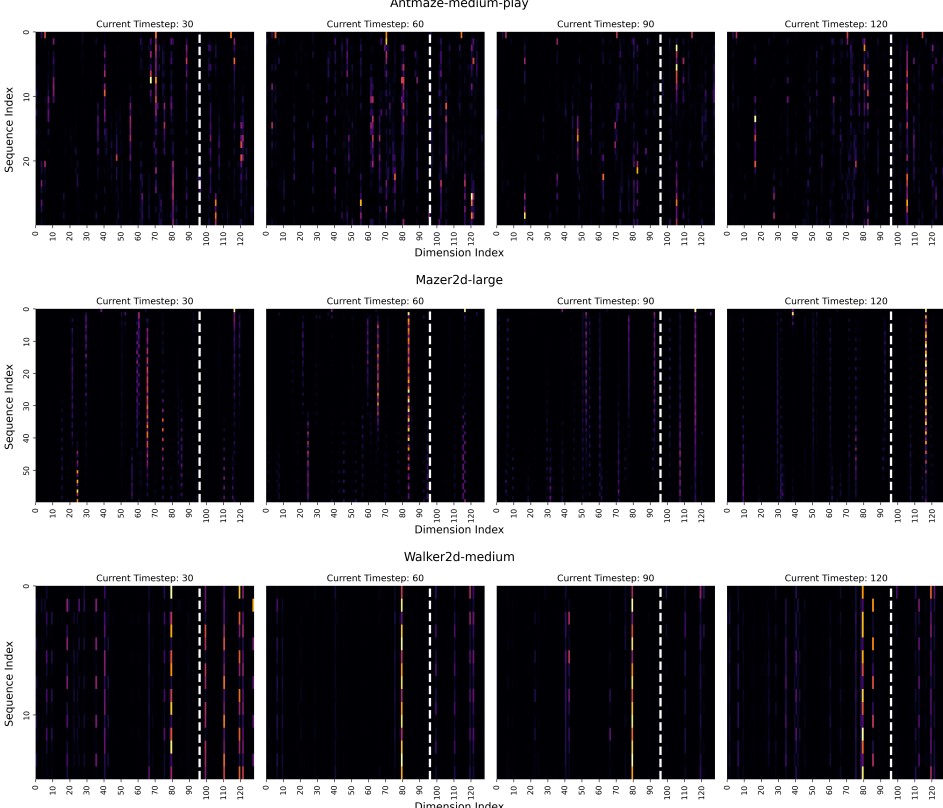

Figure 11: The selective weight matrix $A$ for Antmaze-medium-play, Walker2d-medium, and Maze2d-large at different timesteps. The Dimension ratio $\xi$ for these examples is 75 %.

We visualize the heat map of the selective matrix $A$ in Figure 11, the $X$ axis represents the dimension index, the $Y$ axis represents the timestep, and the model was trained with 128 hidden dimensions, dimension ratio 25% (dimensions 0-95 for long-term, and 96-127 for the short-term branch; dashed line separating both branches). It showcases how this module dynamically assigns weights to dimensions (branches) based on their importance at each timestep. Particularly, the maze2d-large and

walker2d-m datasets, show concentrated weight distributions, meaning only a few dimensions contribute to predicting the current token. In contrast, the heatmap for antmaze shows a more dispersed weight distribution, indicating both local (Markovian) and global (non-Markovian) dependencies are important for decision-making in this task. In walker2d-m, branches' outputs are sometimes competitive (one branch carries the highest weights, see timesteps 60 and 90), while other times they are cooperative (at timestep 30). This demonstrates how HFM emphasizes valuable features while minimizing the influence of less relevant ones from both long-term and short-term branches, thereby enhancing its flexibility to integrate local and global representations.

## B.4   Does LSDT Maintain the Same RTG Conditioning as the Original DT for Trajectory Generation ?

To intuitively demonstrate the impact of conditioning on a specified desired RTG on LSDT's performance, we evaluated it across a broad range and plotted the curve showing the relationship between RTG, and the scores in Figure 12. The yellow band represents the standard deviation of scores. For each environment, we tested the previously trained best model without the hybrid fusion module. This involved evaluating 20 sampled Return-To-Go (RTG) values, with each RTG being tested 100 times to obtain comprehensive results. The best RTG of LSDT is reported in Table 5. Consistent with findings from the DT paper (Chen et al., 2021), we observed that the scores achieved are highly related to the initially conditioned RTG, indicating that our LSDT also well models the distribution of returns. Furthermore, our results indicate that the performance of LSDT tends to reach saturation when conditioned on a sufficiently high score. It is noteworthy that for most tasks, conditioning on a single RTG value seems sufficient to achieve near-optimal scores. This finding is particularly intriguing as it suggests that fine-tuning the RTG as a hyperparameter may not be necessary. For example, we can obtain near-optimal performance in locomotion tasks by conditioning on a single RTG of 7000 yields comparable results. However, this is an initial hypothesis based on current data, and we will further test it across more environments in our future work.

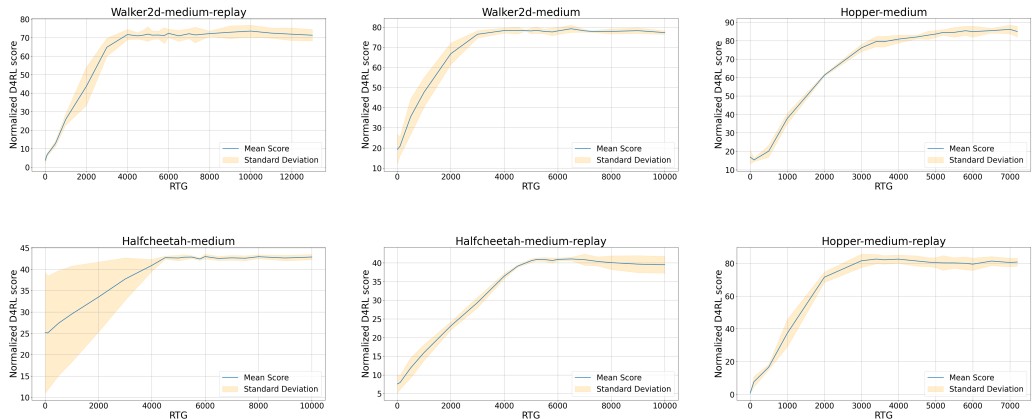

Figure 12: Normalized D4RL scores achieved by LSDT conditioned on 20 sampled desired RTG in Open AI MuJoCo locomotion tasks.

## B.5   Is Our Proposed Goal-State Concatenation the Only Way to Provide Additional Information to DT?

To enhance DT's path planning in environments with diverse goals and sparse rewards, we explored various methods for incorporating the goal position coordinates in AntMaze. Intuitively, we first tried to implicitly incorporate the goal information of each trajectory into each token by adding a learned embedding similar to the time embedding used in DT or position embedding used in NLP. Moreover, we experimented with separately encoding the goal state into each Return-To-Go (RTG), action, and state representation. However, these methods did not yield improvement. It appears that the DT struggled to effectively differentiate between time embedding and goal embedding. This suggests that DT might require more explicit or differently structured goal information to enhance

its decision-making capabilities in complex environments. Therefore, we separately experimented with explicitly concatenating the goal information solely to the state, action, or RTG tokens within the trajectories. Through these trials, we found that concatenating the goal vector with action or RTG hugely decreases DT's performance. This negative effect may be attributed to the constraints imposed by integrating goal information. When goal data is added to action or RTG, it potentially restricts the model's flexibility in stitching trajectories. As a result of these findings, we have selected goal-state concatenation as our preferred approach to enhancing DT's performance. This approach is also aimed at modeling the probability distribution of potential destinations or directions accessible from a given state, thereby providing a better understanding of the state's implications.

### B.6 How does Our Proposed LSDT Influences the Context Length Required to Achieve Optimal Performance Compared to the Context Length in the Standard DT?

The context length is a critical hyperparameter in DT that determines how many past tokens will be considered when we generate the next action. In standard DT and its variant QDT, a context length $K$ of 20 has been shown to yield better performance on OpenAI MuJoCo tasks. However, for our Long-Short Decision Transformer, we observe optimal results with a relatively shorter context length in the majority of environments like Walker2d and Halfcheetah. This suggests that our model efficiently utilizes available information to generate better actions, potentially due to its enhanced capability to capture fine-grained details through CNNs. The context length of LSDT for training and evaluation are shown in Table 5.

Table 5: Context Length and Return-to-go Setting for LSDT.

| Environment | Context length $K$ | Return-to-go conditioning |
|---|---|---|
| Maze2d (umaze, medium, large) | 20 | 300 |
| AntMaze (umaze, medium) | 10 | 100 |
| Walker2d-medium | 5 | 7000 |
| Halfcheetah-medium | 10 | 7000 |
| Hopper-medium | 10 | 7000 |
| Walker2d-medium-replay | 10 | 7000 |
| Halfcheetah-medium-replay | 10 | 7000 |
| Hopper-medium-replay | 20 | 7000 |
| Walker2d-medium-expert | 10 | 10000 |
| Halfcheetah-medium-expert | 10 | 15000 |
| Hopper-medium-expert | 20 | 10000 |

### B.7 Instructions for Selecting Convolution Kernel Size.

In DT and its variants, reinforcement learning problems are formulated as sequence modeling tasks, where future information is unavailable to the model during action generation. Analogous to masked attention in transformers, the weights corresponding to future time steps are masked in the convolution kernel to prevent information leaking. In DT, a unit of input to the attention module consists of three tokens — Return-To-Go (RTG), state (s), and action (a). Our Long-Short LSDT incorporates a dual-branch structure that aligns with this input format. To get better performance, we find that the convolutional kernel size, particularly the part beyond the mask denoted as $k_m$, should also be a multiple of 3, which alignment with the format of the input tokens in the attention branch. The kernel size can be computed as $(k_m * 2) - 1$. In our approach, we use a $k_m$ value of 6 to capture local patterns in adjacent timesteps.

we suggest choosing an appropriate kernel size based on the context length. The Lite Transformer (Wu et al., 2020) and the study introducing dynamic convolution (Wu et al., 2019) both recommend using larger kernel sizes, such as 15 or 31, which we found to be less effective for the masked attention block in our LSDT's decoder. The primary reason is the context length in our model, which is considerably shorter than what is typically used in NLP models. A large kernel size in our context can lead to overfitting and slow down computation. For example, with a context length of 1, we only

deal with three tokens at a time. Choosing a kernel size of 15 in such a scenario would mean that the convolution operation extends well beyond the relevant context, processing a significant amount of masked or irrelevant information. This not only increases the risk of inaccurately modeling dependencies due to introducing noise but also unnecessarily burdens the computational resources.

## B.8 DROPKEY STRATEGY IN LSDT

In the field of computer vision, Li et al. (2022) introduced the Dropkey technique to improve the transformer's capabilities of capturing global information and alleviating overfitting. Unlike the standard attention mechanisms where dropout is applied after computing the attention weights, Dropkey drops the keys in the attention instead of the weights. Intrigued by its potential, we investigated whether this approach could also enhance the performance of LSDT. We adapted this technique by replacing the standard dropout with Dropkey in our attention branch. Our experiments were conducted in the AntMaze environments over 4 random seeds without HFM module. The results in Table 6 indicate that employing DropKey in our LSDT is not only harmful to its performance but also introduces instability. Therefore, we didn't report this strategy in our final model. However, we have retained the DropKey implementation in our code, which is accessible for our further exploration or those interested in it. This includes the possibility of enhancing model performance through a scheduled decrease in the dropout rate, a feature we did not explore in depth, due to its suboptimal performance in our preliminary experiments.

Table 6: Impact of Dropout techniques on LSDT Performance.

| Environment | Standard Dropout | DropKey |
|---|---|---|
| Antmaze-umaze | **84.5 ± 1.7** | 80.25 ± 1.9 |
| Antmaze-umaze-diverse | **81.5 ± 1.7** | 76.0 ± 3.1 |
| Antmaze-medium-play | **71.5 ± 6.2** | 59.3 ± 3.6 |
| Antmaze-medium-diverse | **68.5 ± 5.0** | 50 ± 17.7 |

## B.9 FILTER SIZE EXPERIMENTS

The convolution-only DT achieves strong performance in Markovian datasets due to its ability to capture local dependencies. By setting the kernel size $Ke$ to 6, it effectively models inherent associations between elements in two consecutive timesteps. However, this configuration limits the model's ability to incorporate more historical information, which is essential for learning global patterns in non-Markovian tasks. To investigate whether a larger kernel size can lead to better performance in these tasks, we experiment with the Antmaze-medium-play task on DC in Table 7. The results show that increasing the kernel size provides no further improvement and may even degrade performance.

Table 7: Kernel size influence on Antmaze-medium-play. values are reported over 3 random seeds

| Environment | $Ke$ =6 | $Ke$ =12 | $Ke$ =18 |
|---|---|---|---|
| Antmaze-medium-play | **69.5 ± 3.6** | 52.3 ± 5.9 | 57.0 ± 6.1 |

## B.10 ALTERNATIVE WAYS OF CHANNEL SPLITTING AND INFORMATION ROUTING

In the early stages of our design, we explored various routing mechanisms to enhance the model's expressiveness and improve the overall performance of LSDT. While these alternatives did not outperform the final structure proposed in this paper, exploring more alternative configurations could provide readers with valuable insights for further optimizing our architecture in the future. Here, we detail one of the most promising alternatives (referred to as LSDT-weight), which has the potential to eliminate the need for manually setting the hyperparameter dimension ratio $\xi$. In this configuration, the full embedding is fed into both branches instead of a subset, and the outputs of the branches

are combined through a learnable weighted sum, computed using a modified version of our HFM module. At the suggestion of a reviewer, we also evaluated an alternative routing mechanism, referred to as LSDT-res, which introduces two separate linear projections with residual connections before feeding inputs to the branches. This design aims to enhance feature extraction and potentially improve performance. The performance of these alternatives is reported in Table 8 as the mean and standard deviation over three random seeds, using 20 rollouts for Walker2d-medium and Halfcheetah-medium-replay, and 100 rollouts for Maze2d-medium.

Table 8: Performance comparison across tasks for LSDT-res, LSDT-weight, and LSDT.

| Task | LSDT-res | LSDT-weight | LSDT |
|------|----------|-------------|------|
| Walker2d-medium | $78.3 \pm 3.3$ | $78.4 \pm 0.5$ | $81.0 \pm 0.4$ |
| Halfcheetah-medium-replay | $40.9 \pm 0.6$ | $41.8 \pm 0.3$ | $42.9 \pm 0.3$ |
| Maze2d-medium | $67.6 \pm 3.0$ | $64.0 \pm 4.2$ | $68.4 \pm 3.9$ |

Our preliminary experiments revealed that these designs currently do not result in performance improvements. On the contrary, LSDT-weight increased the total number of parameters for each branch and extended the training time, as both branches processed the full input dimensionality instead of a subset. Nevertheless, its ability to dynamically integrate output features from different branches without relying on a pre-set dimension ratio highlights a potential direction to further improve our LSDT, which warrants further exploration in future research.

## C    EXTENSIVE EXPERIMENTS.

To further verify the performance of the proposed method, we test LSDT on more complex Adroit tasks. The CQL and IQL values are reported from IQL (Kostrikov et al., 2021), and EDAC values from the original paper (An et al., 2021). We report the mean and standard deviation of 5 random seeds, with 20 evaluations each.

As shown in Table 9, the general results are consistent with those from our other evaluations on D4RL benchmarks. LDST performs first or second in most cases, showcasing its effectiveness and adaptability across various RL tasks. The one exception to this is the relocate-cloned-v0, which is a hard task for most offline RL algorithms. Moreover, the performance gap between the '-human' and '-cloned' datasets suggests that LSDT performs better with higher-quality data.

Table 9: Adroit evaluation results.

| Task | CQL | IQL | EDAC | LSDT |
|------|-----|-----|------|------|
| pen-human-v0 | 37.5 | 71.5 | 52.1±8.6 | **109.2±6.2** |
| hammer-human-v0 | 4.4 | 1.4 | 0.8±0.4 | **12.0±6.0** |
| door-human-v0 | 9.9 | 4.3 | 10.7±6.8 | **18.6±8.8** |
| relocate-human-v0 | 0.2 | 0.1 | 0.1±0.1 | **1.5±0.4** |
| pen-cloned-v0 | 39.2 | 37.3 | **68.2±7.3** | 63.6±8.2 |
| hammer-cloned-v0 | 2.1 | 2.1 | 0.3±0.0 | **9.4±4.9** |
| door-cloned-v0 | 0.4 | 1.6 | **9.6±8.3** | 5.0±3.8 |
| relocate-cloned-v0 | -0.1 | -0.2 | 0.0±0.0 | 0.0±0.1 |

## D    EXPERIMENTS DETAILS.

Our code can be found in the submitted material. The Decision Transformer and our Long-Short Decision Transformer, used in the experiments, are constructed based on the official DT open-source code[1] and the min-decision transformer[2]. Additionally, the Dynamic Convolution imple-

---

[1]https://github.com/kzl/decision-transformer
[2]https://github.com/nikhilbarhate99/min-decision-transformer

mented in our study is developed using the Fairseq toolkit[3]. The code for Decision ConvFormer (DC) is sourced from official codebase[4] Each model is trained for $10^5$ updates. **The implementation of our model is publicly available at** https://github.com/WangJinCheng1998/LONG-SHORT-DECISION-TRANSFORMER/tree/main.

## D.1 DT MODEL DETAILS

The architecture of our DT mainly follows the official DT, with the key modification of substituting the ReLU nonlinearity with GeLU. The hyperparameters were fine-tuned as detailed in Table 11. The context length and RTG for the Antmaze experiments are shown in Table 10.

Table 10: Context length and Return-to-go setting for DT.

| Environment | Context length $K$ | Return-to-go conditioning |
|---|---|---|
| AntMaze-umaze | 20 | 1 |
| AntMaze-umaze-diverse | 20 | 1 |
| AntMaze-medium-play | 20 | 1 |
| AntMaze-medium-diverse | 20 | 1 |

Table 11: Hyperparameteters of DT.

| Hyperparameter | Value |
|---|---|
| Number of layers | 3 |
| Number of attention heads | 1 |
| Batch size | 128 |
| Learning rate | $4 \times 10^{-4}$ |
| Grad norm clip | 0.25 |
| Weight decay | $3 \times 10^{-4}$ |
| Embedding dimension | 128 |
| Nonlinearity function | GeLU |
| RTG scale | 1 |
| Dropout | 0.1 |
| Learning rate decay | Linear warmup for first $10^4$ update |

To compare the performance on the 'Antmaze-medium' environments, we evaluate DC using the same architecture that was implemented for the 'Antmaze-umaze' tasks in their paper. The hyperparameters are sourced from their paper 2. Additionally, we conduct evaluations on the Maze2d tasks using DC$h$, which is introduced in the DC paper to enhance the model's long-term credit assignment capability. In the architecture of DC$h$, the final convolution layer is replaced with an attention layer.

## D.2 IMPLEMENTATION DETAILS OF LSDT

In our architecture of LSDT, notably, the sum of the input dimensions of the two branches must equal the embedding dimension, as we distribute different parts of the embedding dimension to each branch for further processing. The hyperparameters of our LSDT for Maze2d, AntMaze, Adroit, and other locomotion tasks are presented in Table 13. The context length and RTG for training and evaluation are shown in the previous section Table 5.

- Embedding dimension: Following the suggestions from the DC paper Wu et al. (2019), we use an embedding dimension of 256 in 'Hopper-medium' and 'Hopper-medium-replay'. We also observed a significant enhancement in performance with this larger dimension. For other environments, we use a dimension of 128. Additionally, we train 'Hopper-medium' without the action information.

---

[3]https://github.com/pytorch/fairseq/tree/master/fairseq
[4]https://github.com/beanie00/Decision-ConvFormer

Table 12: Hyperparameteters of DC on Antmaze and $DC_h$ on Maze2d domain.

| Hyperparameter | Value |
| --- | --- |
| Number of layers | 3 convolution blocks, (DC) |
| | 2 convolution blocks + 1 attention block, ($DC_h$) |
| Number of attention heads | 1, (DC) |
| | 2, ($DC_h$) |
| Batch size | 64 |
| Learning rate | $1 \times 10^{-4}$ |
| Grad norm clip | 0.25 |
| Weight decay | $1 \times 10^{-4}$ |
| Embedding dimension | 128 |
| Nonlinearity function | GeLU |
| RTG scale | 1 |
| Dropout | 0.1 |
| Context length | 8 |
| Learning rate decay | Linear warmup for first $10^4$ update |

Table 13: Hyperparameteters of LSDT.

| Hyperparameter | Value |
| --- | --- |
| Number of layers | 3 |
| Grad norm clip | 0.25 |
| Weight decay | $1 \times 10^{-4}$ |
| Nonlinearity function | GeLU |
| Convolution type | DynamicConv |
| Kernel size | 3 , Maze2d |
| | 6, other environments |
| RTG scale | 1 AntMaze, Maze2d,Adroit |
| | 1000 Locomotion tasks |
| Dropout | 0.25 |
| Fusion method | HFM (Appendix B.3) |
| Dropout for Convolution | 0.2 |
| Learning rate decay | Linear warmup for first $10^4$ updates followed by single-cycle cosine annealing |

- Batch Size: We use 256 in 'Antmaze','Adroit' and 'Maze2d' to stabilize the training process. For locomotion tasks, we use 64.

- Number of attention heads: For more complex tasks such as Antmaze','Adroit' and Maze2d', we utilize 2 heads in the attention branch, with the exception of 'Antmaze-umaze' where we use 1. For other tasks, we use only 1 head.

- Learning rate: We train LSDT with a learning rate of $1 \times 10^{-4}$ in locomotion tasks and 'Antmaze-medium-diverse', and $2 \times 10^{-4}$ in other environments.

- Number of kernels: To reduce the number of parameters, we use the weights-sharing strategy in our model. The number of kernels can be computed as: (Number of dimensions in convolution branch) $\div 4$

The best channel dimension ratio $\xi$ for the convolution branch on each environment is shown in Table 14.

Table 14: The best channel dimension ratio of each task.

| Tasks | Dimension ratio $\xi$ |
|---|---|
| Antmaze-umaze | 50% |
| Antmaze-umaze-diverse | 75% |
| Antmaze-medium-play | 75% |
| Antmaze-medium-diverse | 50% |
| Maze2d-umaze | 50% |
| Maze2d-medium | 12.5% |
| Maze2d-medium | 25% |
| Walker2d-medium | 12.5% |
| Halfcheetah-medium | 50% |
| Hopper-medium | 100% |
| Walker2d-medium-replay | 50% |
| Halfcheetah-medium-replay | 50% |
| Hopper-medium-replay | 100% |
| Walker2d-medium-expert | 12.5% |
| Halfcheetah-medium-expert | 75% |
| Hopper-medium-expert | 75% |

### D.3 LSDC MODEL DETAILS

To demonstrate that other approaches can benefit from our Long-Short architecture, we present LSDC. In LSDC, we maintain the same hyperparameters and architecture as our LSDT, except we replace the dynamic convolution with the type of convolution used in DC.

## E    EXPERIMENTS COMPUTE RESOURCES

To help practitioners in the community evaluate the practical trade-offs of implementing LSDT, we conduct an empirical analysis of the computational efficiency. All models are trained in the Walker2d environment with the same batch size of 64 and a context length of 10. The experiments are run on a desktop with the following configuration: an NVIDIA 4080 GPU, an Intel 13700KF CPU, and 32GB of RAM. We present a detailed comparison across several metrics, including:

- FLOPs, Total parameters comparisons with DT and DC. (Table 15)

- Training time comparisons with single-branch architectures (DT and DC) (Table 15).

- GPU Memory usage analysis across dimension ratios (Table 16).

Table 15: Comparison of Compute Resources across DT, DC, LSDT, and LSDC.

| Metric | DT | DC | LSDT | LSDC |
|---|---|---|---|---|
| Flops (MMAC) | 17.99 | 11.01 | 17.08 | 16.97 |
| Parameters (M) | 1.13 | 0.9 | 1.09 | 1.09 |
| Memory Usage (GB) | 0.1 | 0.1 | 0.11 | 0.1 |
| Training Time (s) | 381 | 364 | 713 | 473 |

LSDT currently requires longer training times, primarily due to the computational overhead of dynamic convolution in the short-term branch. However, our architecture is highly flexible, allowing for improvements through more efficient convolutional computations. As we discussed in Section 4.4 and illustrated in Table 15, replacing dynamic convolution with alternatives, such as those employed in DC, can significantly reduce runtime. This adaptability highlights the potential for further optimization, offering a pathway for future research to balance efficiency and performance.

Table 16: Memory usage of LSDT under different $\xi$ configurations.

| $\xi$ | Memory |
|-------|--------|
| 12.5% | 0.11GB |
| 25% | 0.12GB |
| 50% | 0.12GB |
| 75% | 0.14GB |

## F  LIMITATIONS

While LSDT demonstrates competitive results across various tasks, there are areas for further improvement. Currently, the channel dimension of the input is proportionally split and fed into each branch, and the optimal dimension ratio, $\xi$, may vary across tasks. To address this, we conducted experiments to identify appropriate $\xi$ values for our implementations. Additionally, in Appendix B.10, we introduce a potential alternative architecture that eliminates the need for manual tuning. Although this approach is still exploratory, it shows promise, and we plan to further investigate how to efficiently and dynamically select $\xi$ in future research.

