# OpenReview forum: "Long-Short Decision Transformer: Bridging Global and Local Dependencies for Generalized Decision-Making"
_ICLR.cc/2025/Conference — ICLR 2025 Poster_

### Official Review · Reviewer_NBtN · 2024-11-02

**Soundness:** 3
**Presentation:** 4
**Contribution:** 2
**Rating:** 6
**Confidence:** 3

**Summary:**

This submission presents a novel architecture for the decision transformer named LSDT that tries to capture both long-term and short range dependencies, such as local spatiotemporal associations, for offline RL tasks. This is achieveed by adding a second, convolutional block next to the self attention block, and the sizes of the two blocks (eg in terms of dimensions) can vary according to the needs of the task to solve.  The paper further proposes to incorporating goal states into the conditioning information for further performance gains in some cases. Experiments are conducted on the D4RL benchmark, where LSDT achieces performance comparable to state-of-the-art RL methods.

**Strengths:**

S1) The motivation if the proposed approach is clear and the paper is well written. The closest related works are cited and the contributions pf the paper are clearly presented and illustrated both qualitatively and quantitatively.
S2) Although the technical contributions are incremental, the paper presents clear figures that illustrate the strenghts of the proposed method, and also ablations that study the key points of the proposed architecture
S3) The method seems to achieve results comaprable to the state of the art.

**Weaknesses:**

W1) The technical contribution is following other works that propose two-branch transformer blocks: appending a convolutional branch in the architecture, and using a straighforward attention masking scheme to capture long vs short dependences.

W2) The split across the two branches is simple, ie across the channels dimension, and it is manual via hyper parameter \xi, an adjustment which is non trivial and highly task based. The submission would be stronger if an automatic way of selecting \xi was proposed. It would be great if the authors discussed potential methods for automating this.

W3) An issue uncovered in Fig7 is the following: this paper is presenting a method for a compromize between DT and CDT, via splitting the dimensions across the two. However, we see in Fig 7 that in most cases there doesnt seem to be a "peak" inbetween the already existing edge cases.  Could the authors discuss this further? It would be nice to address this apparent discrepancy between one of the motivationsof their method and the results shown.

**Questions:**

Q1) Could the CDT subfig in Figure 1 also be depicting the Decision Convformer (DC)? It would be clearer if DT and DC were clearly marked if this is  the case

Q2) How well would the DC perform if one substitutes their conv filters with DynamicConv?

Q3) Are \xi identical across all datasets in Table 1?

Q4) Is goal conditioning used in Table 1, and if not what is the performance of LSDT-gs in this case?

Q5) IS FIg3 depicting a  single sequence or averages over many?

---

> ### Author Response · Authors · 2024-11-19
>
> Thank you for your thoughtful review of our research. We appreciate your positive remarks about the effectiveness of our designs. We hope this individual response will answer your concerns in detail：
>
> >**Q**: "Could the CDT subfig in Figure 1 also be depicting the Decision Convformer (DC)? It would be clearer if DT and DC were clearly marked if this is the case"
>
> **A**: Thank you for this suggestion. In Figure 1, we present the overall structure of three distinct types of Decision Transformers (DT). The Decision Convformer (DC) is included within our classification of Convolution-only DT (CDT) as it replaces the attention mechanism with a convolution module. To improve clarity, we will revise the label in Figure 1 from “Convolution-only DT (CDT)” to “Convolution-only DT (CDT, e.g., DC)” to help readers better understand our categorization and motivation.
>
> >**Q**: "How well would the DC perform if one substitutes their conv filters with DynamicConv?"
>
> **A**: DC keeps most of the settings as DT except for replacing the standard masked attention block with causal convolution filters. In Figure 7, we illustrate the scenario you're referring to: when $\xi$ equals 100%, our LSDT effectively becomes a Dynamic Convolution-only DT, equivalent to substituting DC’s convolution filters with DynamicConv. Under these conditions, it still shows improvement compared with the original DT on most tasks.
>
> >**Q**: "Are $\xi$ identical across all datasets in Table 1?"
>
> **A**: To provide sufficient experimental details, we have listed all hyperparameter values $\xi$ for each dataset. Please refer to Appendix D.2 Table 12.
>
> >**Q**: "Is goal conditioning used in Table 1, and if not what is the performance of LSDT-gs in this case?"
>
> **A**: Yes, we use goal conditioning for the Antmaze tasks, as mentioned in lines 397–401. We will further clarify this in the title of Table 1 for better clarity.
>
> >**Q**: "Is Fig3 depicting a single sequence or averages over many?"
>
> **A**: As a demonstration, Figure 3 presents the heatmap for a single sequence, highlighting the associations captured by different models and branches. During the inference stage, the specific attention scores vary based on the input provided. Since each rollout involves a sequence of unique inputs, resulting in distinct attention scores, averaging these scores across sequences would blur the context-specific dependencies they represent. We will clarify this in the title of Figure 3.
>
> >**Q**:" The submission would be stronger if an automatic way of selecting $\xi$ was proposed. It would be great if the authors discussed potential methods for automating this."
>
> **A**: We appreciate your thoughtful observation about automatically setting $\xi$ and we recognize that as a potential improvement direction mentioned in Appendix F (Limitation section) for future work. The dimension ratio is a hyperparameter, similar to the Context Length in Decision Transformers, which also requires task-specific tuning. However, beyond conducting ablation studies on task-based hyperparameters like context length (as done in the DT paper), we provide practical guidelines specifically for the dimension ratio $\xi$ that we introduced. These guidelines, based on extensive experimentation, are intended to help users apply our algorithm more effectively and reduce tuning costs (see Section 4.5, lines 473–485).
>
>
> Moreover, we agree that discussing potential methods for automating this would be great. While automating $\xi$ selection is beyond the current scope, the extended exploration and experiments presented in the Appendix provide a strong foundation for such future advancements. One possible approach is to feed the entire hidden features into both branches, instead of splitting them, and combining the outputs with a dynamic and selective weighted sum. The weight matrix for each branch’s output can be generated by our optional module (HFM) as described in Appendix B.3. However, a potential drawback is that this approach will increase the model’s total parameters and training costs, as each branch processes the full hidden dimension rather than only a part. Although this approach remains exploratory, we believe this offers us a potential way to avoid the need for manual tuning of the dimension ratio $\xi$. We will include the above discussion in Appendix F, while keeping the primary focus of the paper on demonstrating the effectiveness of our long-short architecture for offline RL.

---

> ### Author Response · Authors · 2024-11-19
>
> >**Q**:"we see in Fig 7 that in most cases there doesnt seem to be a "peak" inbetween the already existing edge cases. Could the authors discuss this further? It would be nice to address this apparent discrepancy between one of the motivations of their method and the results shown."
>
> **A**: This is an interesting observation that does not undermine the motivations of our method, as discussed in Section 4.5. Our LSDT consistently demonstrates strong performance across both Markovian and non-Markovian datasets. For Markovian datasets, the performance of DT is enhanced by integrating a local feature extractor into DT across all dimension ratios in most cases (lines 461–462). Notably, a performance ‘peak’ or plateau occurs at a dimension ratio of 12.5% in Figure 7, supporting the motivation behind DC and one of the motivations of our LSDT: adding a short-term branch enhances DT’s performance on Markovian tasks. For non-Markovian datasets, such as maze2d, the performance ‘peak’ is also achieved by our LSDT compared to (long-term only) DT and (short-term only) CDT. Based on the performance plateau observed across several tasks, we conclude that selecting a dimension ratio of 50% achieves suboptimal yet acceptable performance on most datasets (see lines 485). We will further clarify this discussion in Appendix B.
>
>  The key takeaway here is that our architecture is flexible enough to use a wide range of approaches for either branch, further tuning it to the specific requirements at hand. In any case, our approach will contain a dedicated branch/approach to deal with local and global features, allowing it to deal with diverse tasks.
>
>
>  We hope our response addressed your main concerns, and thanks again for your suggestions. We are really excited to see how these changes will greatly improve the paper and look forward to working with you towards that during the discussion phase.

---

> > ### Author Response · Authors · 2024-11-25
> > **Willing to clarify further concerns you may have**
> >
> > Dear reviewer NBtN, we appreciate your valuable feedback and insights. We have addressed your questions and concerns in detail, and carefully revised our work accordingly. We would greatly appreciate it if you could review our responses and let us know if they address your concerns. If you have any further suggestions or comments, we would be delighted to engage in discussion. Thank you once again for your time and support!

---

> > > ### Comment · Reviewer_NBtN · 2024-11-26
> > > **Thank you for the responses**
> > >
> > > Thank you for all the clarifications and answers.
> > >
> > > > we have listed all hyperparameter values  for each dataset.
> > > The fact that optimal \ksi varies so much (in Table 13), shows that the method indeed needs per-dataset tuning.
> > >
> > > >  we conclude that selecting a dimension ratio of 50% achieves suboptimal yet acceptable performance on most datasets
> > > I get your argument that one can find a middle ground with your method that works more-or-less well in many cases, however, if one knows that the peak is most likely on either the one or the other edge case, then, in practice, it seems to be better to test these 2 options for each case.
> > >
> > > Thanks again!

---

> > > > ### Author Response · Authors · 2024-11-27
> > > >
> > > > Thank you again very much for your thoughtful review and thank you for your reply!
> > > >
> > > > >The fact that optimal \ksi varies so much (in Table 13), shows that the method indeed needs per-dataset tuning
> > > >
> > > > We agree that achieving optimal results requires tuning the dimension ratio, similar to other hyperparameters such as context length. Following your suggestion, we are happy to explore alternatives to automate this process. We have updated the paper to include this discussion and have also incorporated it into our future research plans.
> > > >
> > > > >if one knows that the peak is most likely on either the one or the other edge case, then, in practice, it seems to be better to test these 2 options for each case
> > > >
> > > > We really appreciate your suggestion. We propose using 50% as a generally acceptable choice for practical applications without dimension ratio tuning, particularly when prior knowledge of the dataset type is unavailable. This provides a balanced approach that effectively manages the trade-off between hyperparameter tuning cost and performance. We will clarify this in Section 4.5 (third guideline). However, one key advantage of our approach is that it can pivot between using branches, if the task demands so. If further information about the task suggests that one edge case is more favorable, the model can be switched to a certain configuration without losing generality (e.g., 100% for Halfcheetah-medium and 0% for Maze2d-medium). This flexibility ensures that our method is not strictly tied to the middle ground (e.g., 25%,50%,75%) but can also pivot toward task-specific extremes (0%, 100%) when necessary.
> > > >
> > > >
> > > > We have updated the PDF accordingly and are willing to incorporate any further insightful suggestions into the camera-ready version. Thank you so much for your constructive and constructive suggestions, which have helped improve the clarity of our work and reaffirmed its potential impact on advancing practical applications in this area.

---

### Official Review · Reviewer_vshZ · 2024-11-04

**Soundness:** 3
**Presentation:** 2
**Contribution:** 2
**Rating:** 6
**Confidence:** 2

**Summary:**

The manuscript proposes the Long-Short Decision Transformer (LSDT), a model that effectively combines the strengths of Decision Transformer (DT) and Decision Convformer (DC) through a dual-branch architecture.
This architecture uses a dimension ratio, ξ, to dynamically control the proportion of channels processed by each branch—self-attention for long-range dependencies and convolution for local patterns.
The approach is simple yet intuitive, as it leverages complementary feature extraction methods to enhance both global and local dependency modeling, adapting flexibly to diverse reinforcement learning tasks.

**Strengths:**

The proposed Long-Short Decision Transformer (LSDT) presents an interesting approach by combining attention and convolutional pathways to handle both global and local dependencies effectively.
This dual-branch design not only enhances interpretability but also achieves strong performance across diverse reinforcement learning tasks, particularly outperforming benchmarks on the D4RL dataset.
The flexibility offered by the dimension ratio, ξ, adds adaptability, allowing LSDT to maintain robust performance in both Markovian and non-Markovian environments.
Additionally, the model’s simplicity and modularity enable straightforward integration into existing architectures, further extending its practical utility.

**Weaknesses:**

The experiments on the routing mechanism in the LSDT are limited, leaving open questions about potential improvements.
Given that the proposed method can be viewed as a two-expert MoE (Mixture of Experts) architecture, alternative ways of channel splitting and information routing could be explored.
For instance, introducing two separate linear projections with residual connections before feeding inputs to the branches could enhance feature extraction.
Another approach might involve feeding the same embedding into both branches and combining the outputs with a learnable weighted sum.
Evaluating these alternative configurations could offer insights into optimizing the dual-branch structure, potentially enhancing the model’s performance and efficiency.

**Questions:**

See weakness

---

> ### Author Response · Authors · 2024-11-19
>
> Thank you for your thoughtful review of our research and positive remarks about the flexibility and effectiveness of our design. We appreciate your constructive feedback about potential improvements in our LSDT and hope our extended experiments in this response will address your concerns in detail:
>
> >**Q**: "Introducing two separate linear projections with residual connections before feeding inputs to the branches could enhance feature extraction. Another approach might involve feeding the same embedding into both branches and combining the outputs with a learnable weighted sum. Evaluating these alternative configurations could offer insights into optimizing the dual-branch structure, potentially enhancing the model’s performance and efficiency. "
>
> **A**:In the early stages of our design, we explored various routing mechanisms to enhance the model’s expressiveness and improve the overall performance of LSDT. Notably, we also considered the second idea you proposed (referred to as LSDT-weight)—feeding the same embedding into both branches and combining the outputs using a learnable weighted sum generated by a modified version of our HFM module. It is gratifying to see that our thoughts align on this approach, as this idea was precisely the motivation behind designing the optional HFM module.
>
> While this approach avoids the need for manually tuning the dimension ratio by multiplying a learned weight matrix to each branch separately to dynamically select the important channels and combine the outputs, our experiments showed that it did not lead to performance improvements. Instead, it increased the total parameters of each branch and extended the training time, as both branches processed the full dimensionality of the input rather than a subset of it.
>
> We agree with your thoughtful suggestion that exploring more alternative configurations could offer readers more valuable insights for further optimizing our architecture in the future, potentially improving model performance and eliminating the need for manually setting the hyperparameter $\xi$. Therefore, we are willing to conduct a preliminary evaluation comparing our LSDT with the approaches you suggested. We report the mean and standard deviation of 3 random seeds, using 20 rollouts for Walker2d-medium and Halfcheetah-medium-replay, and 100 rollouts for Maze2d-medium. For clarity, we refer to your first proposed method as LSDT-res and the second as LSDT-weight.
>
> | Task                         | LSDT-res       | LSDT-weight    | LSDT          |
> |------------------------------|----------------|----------------|---------------|
> | **Walker2d-medium**          | 78.3 ± 3.3     | 78.4 ± 0.5     | 81.0 ± 0.4    |
> | **Halfcheetah-medium-replay**| 40.9 ± 0.6     | 41.8 ± 0.3     | 42.9 ± 0.3    |
> | **Maze2d-medium**            | 67.6 ± 3.0     | 64.0 ± 4.2     | 68.4 ± 3.9    |
>
> As shown in the table, LSDT achieves better performance compared to the two alternative designs, reaffirming the effectiveness of our routing design choice. While these alternatives may not currently achieve better performance, evaluating them could inspire future research, as it is possible that an alternative approach or routing mechanism could surpass our design in specific scenarios. We will include these additional experiments in Appendix B.10 to encourage further exploration.
>
> We hope our response has addressed your main concerns, and we sincerely thank you again for your valuable suggestions. We are excited to see how these changes will significantly enhance the paper and look forward to engaging with you further during the discussion phase.

---

> > ### Author Response · Authors · 2024-11-25
> > **Willing to clarify further concerns you may have**
> >
> > Dear reviewer vshZ, we appreciate your valuable feedback and insights. We have addressed your concerns in detail, conducted analysis and additional evaluations on alternative configurations you suggested, and carefully revised our work accordingly. We would greatly appreciate it if you could review our responses and let us know if they address your concerns. If you have any further suggestions or comments, we would be delighted to engage in discussion. Thank you once again for your time and support!

---

> > > ### Comment · Reviewer_vshZ · 2024-11-28
> > > **No further concerns**
> > >
> > > Thank you for your thorough rebuttal.
> > > All my concerns have been addressed.
> > > Unfortunately, after careful consideration, I will be unable to raise my score, and I would like to keep my preliminary rating.

---

### Official Review · Reviewer_wr6s · 2024-11-10

**Soundness:** 3
**Presentation:** 3
**Contribution:** 3
**Rating:** 6
**Confidence:** 4

**Summary:**

This paper targets two critical limitations in existing Decision Transformers (DTs)-based offline reinforcement learning: (1) DTs' weakness in capturing local dependencies which is associated with Markovian properties in many offline-RL datasets; (2) DTs’ struggles with goal-reaching tasks involving diverse goals and sparse rewards. To address these two issues, the authors propose a dual-branch architecture, Long-Short Decision Transformer (LSDT), that combines self-attention for global patterns and dynamic convolution for local features, along with a goal-state conditioning mechanism. The main contribution is the asymmetric dimension-splitting between branches. Experiments are conducted across diverse environments and the results demonstrate state-of-the-art performance on several D4RL benchmarks, particularly showing significant improvements in complex environments like Antmaze-medium and Maze2d-large.

**Strengths:**

**(S1) The Significance and Clear Presentation of Research Question:**
The paper addresses a critical challenge in Transformer-based offline-RL with great clarity and motivation. The key motivation lies in identifying and solving the global and local dependencies modeling, which has been a critical limitation in existing RL architectures. Unlike existing approaches that treated this as an either-or choice (DTs focusing on global, DCs on local dependencies), the proposed LSDT introduces dual-branch parallel processing that captures both types of dependencies simultaneously, and thereby hope to benefit offline-RL tasks.

**(S2) Technical Soundness:**
The technical contributions of the proposed LSDT are well-reasoned. Concretely, the Dynamic Convolution in the short-term branch represents an improvement over static convolution methods. By dynamically generating weights based on current input rather than using fixed weights after training, the model may achieve better generalization across diverse environments. The connection between LSDT and the targeted question is clear and reasonable. This is evident in the experiments where LSDT achieves great performance in both Markovian environments (with strong local feature requirements) and the non-Markovian ones (with strong long-range modeling requirements). The asymmetric dimension splitting, controlled by ratio ξ, provides flexibility in adapting the architecture to different tasks. This is demonstrated through ablation studies showing how different ratio values optimize performance across various environments (e.g., ≥50% for Markovian and ≤50% for non-Markovian tasks). Complementing these components, the goal-state conditioning mechanism, while straightforward, addresses the path planning limitations in sparse reward settings. The results on Antmaze tasks (showing improvements from 0% to over 70% success rate) provide strong validation of this method. The optional Hybrid Fusion Module, though not always necessary, presents an approach to dynamic feature selection, with visualization studies clearly showing its ability to weight different features based on task requirements adaptively.

**(S3) Thorough Experiments and Ablation Studies:**
The experiments are thorough and well-structured. The authors conduct extensive experiments across diverse environments, including MuJoCo locomotion tasks, Maze2d navigation, AntMaze environments, and Adroit manipulation tasks. The ablation studies are comprehensive, examining the impact of different architecture choices, such as dimension ratio (across 20 different values), kernel size (from 6 to 18), and fusion methods. The performance gains are consistently shown across both Markovian datasets (achieving 82.0 average normalized score in locomotion tasks) and non-Markovian datasets (significant improvements in Maze2d-large and Antmaze-medium). The authors also provide an empirical analysis of failure cases and performance characteristics under different hyper-parameter settings, making the results valuable for practical implementation and future studies in the community.

**Weaknesses:**

**(W1) Limited Theoretical Analysis and Foundation:**
While the empirical results are impressive, the lack of formal analysis leaves several crucial questions unanswered. First, there is limited theoretical justification for why the combination of self-attention and dynamic convolution in parallel branches should perform better than sequential or alternative operations in RL. Second, the interaction between the two branches lacks theoretical examination. For instance, how information flows between them and potential interference effects, remain unexplored. Third, the paper provides no theoretical bounds or guarantees on the optimal dimension ratio ξ. While empirical guidelines are provided (e.g., ≥50% for Markovian tasks), a theoretical framework supporting these choices would make the approach more principled.

**(W2) Limited Technical Originality and Domain-Specific Analysis:**
While the proposed dual-branch architecture exhibits strong empirical results, its design principle of combining global and local feature extraction has been extensively explored in other domains, particularly Computer Vision and NLP. As such, this work would significantly benefit from a more in-depth analysis of why and how this architecture specifically addresses the unique challenges in reinforcement learning. For example, a deeper analysis to show the relationship between Markovian/non-Markovian properties and global-local feature extraction (how local convolution patterns capture Markovian state transitions, while global attention mechanisms model longer-term dependencies in policy behavior). This could be supported by visualization that expresses how different types of dependencies are processed in each branch for decision-making.

**(W3) Architecture Designs and Computational Efficiency Concerns:**
Several architectural choices raise concerns about practical applicability and scalability. The requirement for manual tuning of the dimension ratio ξ represents a fundamental limitation in the adaptability of LSDT architecture. While the authors provide general guidelines (≥50% for Markovian, ≤50% for non-Markovian tasks), the absence of an adaptive way for determining optimal ratios necessitates extensive experimentation for each new environment. This becomes particularly problematic in real-world applications where task characteristics may not be clearly Markovian or non-Markovian or may shift during deployment. Additionally, the computational complexity of the dual-branch architecture deserves deeper analysis. While the authors mention reduced context length requirements compared to standard DT, they don't provide a comprehensive analysis of the compute-performance trade-off.

**Questions:**

**(Q1)** Could the authors provide training curves showing the convergence behavior of both branches? This analysis would provide insights into how the dual-branch architecture learns to balance global and local features during training. Specifically, it would be valuable to see:

- The loss trajectories for each branch separately;
- The relative contribution of each branch to the final decision at different training stages;
- Whether one branch tends to converge faster than the other;

**(Q2)** Have the authors considered using adaptive dimension ratios that change during training?

**(Q3)** To better understand the architecture benefits, I suggest the authors provide t-SNE plots of learned state representations from both branches. This could provide more insights into the technical contribution of this work.

**(Q4)** I suggest the authors conduct an empirical analysis of the computational efficiency. This would help practitioners in the community evaluate the practical trade-offs of implementing LSDT, including:

- FLOPs comparison with baseline models;
- Memory usage analysis for different dimension ratios;
- Training time comparison with single-branch architectures;

---
**Additional Comments:**

I hope my review helps to further strengthen this paper and helps the authors, fellow reviewers, and Area Chairs understand the basis of my recommendation. I also look forward to the rebuttal feedback and further discussions, and would be glad to raise my rating if thoughtful responses and improvements are provided.

---
## **-------------------- Post-Rebuttal Summary --------------------**

The additional experiments, discussions, and revised manuscript provided by the authors have significantly strengthened the work and addressed most of my concerns. I suppose this work can provide knowledge advancement to the community, and I look forward to the final version manuscript, which incorporates the additional insights and information presented in the rebuttal stage.

---

> ### Author Response · Authors · 2024-11-19
>
> Thank you for your thoughtful review of our research. We appreciate your constructive feedback and hope our suggested changes and this individual response will address your concerns in detail:
>
> >**Q**: “Could the authors provide training curves showing the convergence behavior of both branches? Specifically, it would be valuable to see: The loss trajectories for each branch separately; The relative contribution of each branch to the final decision at different training stages; Whether one branch tends to converge faster than the other; ”
>
>  **A**: We understand there might have been a misunderstanding regarding the training phase of our proposed architecture and we will try to clarify this. As noted in lines 237–238, LSDT retains most of the settings used in DT, including the same training and inference stage, with the main differences lying in its internal architecture. For further clarity, we will mention that we are using the same method as DT and DC to train our model in line 238 to ensure it is as clear as possible.
>
> That is, like standard Decision Transformer architectures, we train our LSDT model in a single unified training stage. Our approach modifies the Transformer structure by replacing the masked self-attention block with our proposed masked long-short block, yet the entire architecture is still trained as an integrated whole. Therefore, (1) our model only has a single training curve, as shown in Appendix B.3.2, Figure 10, which provides the training curve for the Walk2d task as an example. (2) LSDT only has one training stage. (3) two branches have the same convergence behavior.
>
> >**Q**:"Have the authors considered using adaptive dimension ratios that change during training?"
>
> **A**: We appreciate your thoughtful observation about using adaptive dimension ratio $\xi$ and we recognize this as a potential improvement in Appendix F (Limitation section). We use the dimension ratio as a hyperparameter, similar to the Context Length in Decision Transformers, which also requires task-specific tuning. Our ablation studies on such task-based hyperparameters like context length (as done in the DT paper), allow us to illustrate their effects, as well as to provide practical guidelines specifically for the dimension ratio $\xi$ that we introduced. Derived from extensive experimentation, these guidelines aim to help users apply our algorithm more effectively and reduce tuning costs.
>
> We recognize that adaptively adjusting the dimension ratio during training is not feasible, as changing the ratio would alter the input feature dimensions for each branch, resulting in a variable parameter count that compromises model stability and consistency. To address this, our proposed HFM module in Appendix B.3 provides a basis for approaches that eliminate the need for a fixed dimension ratio. One possible approach could be to input the full set of hidden features into both branches without splitting, then combine the outputs with a dynamic, selective weighted sum. The weight matrix for each branch’s output can be generated by the modified HFM module. However, a drawback of this method is that it would increase the model’s total parameters and training costs, as each branch processes the full hidden dimension. While this approach is still exploratory, we believe it offers a promising path to avoid manual tuning of the dimension ratio $\xi$. We will include this discussion in Appendix F (Limitations) to inspire further improvements on LSDT, while keeping the paper’s primary focus on demonstrating the effectiveness of our long-short architecture for offline RL.

---

> ### Author Response · Authors · 2024-11-19
>
> >**Q**:To better understand the architecture benefits, I suggest the authors provide t-SNE plots of learned state representations from both branches. This could provide more insights into the technical contribution of this work.
>
> **A**: Thank you for the suggestion. While we appreciate the potential insights that t-SNE plots could provide regarding the learned representations, analyzing the attention scores of each branch may offer a more direct understanding of the relationship between Markovian/non-Markovian properties and global-local feature extraction. As illustrated in Section 3.1, we conducted experiments on non-Markovian and Markovian datasets to intuitively visualize the contributions of each branch in focusing on distinct features or dependencies. In Figure 3 c,d, the heatmaps illustrate how local convolution patterns capture Markovian state transitions, while global attention mechanisms model longer-term dependencies in decision-making. Moreover, to provide a more in-depth of our architecture in addressing RL problems, we demonstrate the impact of the dimension ratio $\xi$ on dependency modeling within each branch for decision-making in Appendix B.2.
>
>
> In Figure 11 (Appendix B.3.2), we present a visualization method to analyze the contributions of different channels and branches in decision-making. This method enables us to determine which channels and branches contribute the most critical information for decision-making, providing a clearer insight into the contribution of each branch. By illustrating how the two branches interact and complement each other, this approach facilitates a more focused interpretation of the model’s technical contributions. We focused on more task-relevant analyses in this work but, should the reviewer panel find this necessary, we would be happy to also include t-SNE plots, adding results to future studies or supplementary materials where deemed suitable.
>
> >**Q**:" I suggest the authors conduct an empirical analysis of the computational efficiency. This would help practitioners in the community evaluate the practical trade-offs of implementing LSDT, including: FLOPs comparison with baseline models; Memory usage analysis for different dimension ratios; Training time comparison with single-branch architectures."
>
> **A**:We agree that offering readers more empirical analysis would be valuable for evaluating the practical trade-offs involved in implementing LSDT. In Appendix E, we included a preliminary analysis of the computational resources required by our LSDT model. To further enhance understanding of our model’s efficiency, we will expand Appendix E to include additional performance metrics, specifically:
>
> - FLOPs, Total parameters comparisons with DT and DC.
> - Memory usage analysis across dimension ratios (12.5%, 25%, 50%, 75%), excluding 0% and 100%, where the model reverts to DT and CDT, respectively.
> - Training time comparisons with single-branch architectures (DT and DC).
>
> The configuration of our experimental platform is provided in Appendix E. All models are trained in the Walker2d environment with the same batch size of 64 and a context length of 10.
>
> | Metric              | DT     | DC     | LSDT   | LSDC   |
> |---------------------|--------|--------|--------|--------|
> | **Flops (MMAC)**    | 17.99  | 11.01  | 17.08  | 16.97  |
> | **Parameters (M)**  | 1.13   | 0.9    | 1.09   | 1.09   |
> | **Memory Usage (GB)**| 0.1    | 0.1    | 0.11   | 0.1    |
> | **Training Time (s)**| 381    | 364    | 713    | 473    |
>
>  | Configuration         | Memory Usage |
> |-----------------------|--------------|
> | LSDT (ξ=12.5%)       | 0.11GB       |
> | LSDT (ξ=25%)         | 0.12GB       |
> | LSDT (ξ=50%)         | 0.12GB       |
> | LSDT (ξ=75%)         | 0.14GB       |
>
> LSDT currently requires longer training times, primarily due to the computational overhead of dynamic convolution in the short-term branch. However, our architecture is highly flexible, allowing for improvements through more efficient convolutional computations. As we discussed in Section 4.4 and illustrated in the table above, replacing dynamic convolution with alternatives, such as those employed in DC, can significantly reduce runtime. This adaptability highlights the potential for further optimization, offering a pathway for future research to balance efficiency and performance.
>
> We hope our response addressed your main concerns, and thanks again for your suggestions. We are excited about how these discussions and improvements can further enhance the quality of this paper, and we look forward to collaborating with you during the discussion phase.

---

> > ### Author Response · Authors · 2024-11-25
> > **Willing to clarify further concerns you may have**
> >
> > Dear reviewer wr6s, we appreciate your valuable feedback and insights. We have addressed your questions and concerns in detail, conducted additional analysis of computational efficiency, and carefully revised our work accordingly. We would greatly appreciate it if you could review our responses and let us know if they address your concerns. If you have any further suggestions or comments, we would be delighted to engage in discussion. Thank you once again for your time and support!

---

> > > ### Comment · Reviewer_wr6s · 2024-11-30
> > > **Official Response from Reviewer wr6s to the Rebuttal**
> > >
> > > Dear Authors,
> > >
> > > I have thoroughly gone through the authors' responses and carefully examined all the additional experimental results provided in the rebuttal. The authors have effectively addressed most of my initial concerns and questions through extra experiments and insightful discussions. In addition, I am impressed by both the comprehensiveness of the responses and the authors’ positive attitude to addressing the concerns raised in my initial review.
> > >
> > > The clarifications have significantly strengthened the paper in several key aspects. First, regarding the training methodology, the authors effectively explained how LSDT implements a unified training approach analogous to standard DT, with simultaneous training of both branches. The training curves in Appendix B.3.2 provide compelling evidence of this unified convergence behavior. Second, discussions of the dimension ratio adaptation challenge are particularly enlightening. The technical constraints outlined, coupled with the proposed HFM module solution, demonstrate thoughtful consideration of both current limitations and future improvements. I strongly recommend the authors present this in the main text, which could provide valuable insights to the audience in the community.
> > >
> > > The additional computational efficiency analysis in Appendix E represents another significant enhancement. The comparison of FLOPs, parameters, memory usage, and training times across different architectures and dimension ratios provides practitioners with valuable insights for implementation decisions. Furthermore, the visualization methods presented in Figure 11 (Appendix B.3.2) effectively illuminate the internal mechanics of the model, clearly demonstrating how different channels and branches contribute to decision-making.
> > >
> > > ---
> > >
> > > While the additional experimental results and clarifications have substantially increased my confidence in LSDT's technical soundness and practical utility, I would like to suggest two points for further improvement in the ultimate revised manuscript:
> > >
> > > - **Research Focus of the Paper:** From my perspective, I recommend reframing the manuscript to emphasize the underlying challenges that LSDT addresses, particularly the distinct requirements of **Markovian and non-Markovian properties in RL** compared to general applications. This shift from a method-oriented to a **research question-oriented** paper presentation would better highlight the significance of the proposed architectures for the RL community.
> > > - **Documentation Completeness:** I strongly encourage incorporating all the additional discussions and experiments from the rebuttal into the final revised manuscript, including the t-SNE visualization in the appendix. These additions would provide valuable insights to the broader research community.
> > >
> > > Based on the thorough responses and the clear potential for these final improvements, I decided to first update my rating from 5 to 6. Overall, this work presents a well-reasoned and effective approach to combining global and local dependencies in RL decision transformers, supported by comprehensive empirical validation and practical implementation insights. The suggested revisions would further strengthen what is already a valuable contribution to the ICLR community.
> > >
> > > I look forward to further discussions, particularly regarding my two suggestions for improvement.
> > >
> > > Best regards,
> > >
> > > Reviewer wr6s

---

> > > > ### Author Response · Authors · 2024-12-01
> > > >
> > > > Thank you for your appreciation of the comprehensiveness of our response and your positive remarks on the contribution of this work to the field! Our discussion of the two further suggestions for improvement you raised are as follows:
> > > >
> > > > > "reframing to better emphasize the underlying research challenges, especially regarding Markovian and non-Markovian properties in RL"
> > > >
> > > >  Following suggestions, we have carefully considered your suggestion and are willing to further emphasize the underlying challenges that LSDT addresses, as outlined below:
> > > >
> > > >  - Introduction Section: " The limitations observed in DT and DC highlight the importance of a model's ability to capture both local and global dependencies, which is crucial for enabling agents to learn an optimal policy from diverse datasets. Typically, in offline RL, datasets are often composed of trajectories generated by various policies, leading to data that is not strictly Markovian and may instead exhibit non-Markovian or mixed characteristics." (Add this to line 69).
> > > >  - Motivation Section: "This is particularly important in offline RL tasks, where models must effectively handle both Markovian (short-term) and non-Markovian (long-term) dependencies present in diverse datasets" (Add this to line  158).
> > > >  - Conclusion Section: "In this paper, we addressed the challenge of effectively handling both Markovian and non-Markovian properties in offline RL datasets by introducing the Long-Short Decision Transformer (LSDT)," (slightly modified the description in lines 526-527).
> > > >  - Experiment Section: Section 4.5 already includes a detailed discussion on the distinct dimension ratio $\xi$ requirements for Markovian and non-Markovian properties in our LSDT. This will, along with other adjustments, form a cohesive narrative.
> > > >
> > > > It is important to note that while making these revisions, we ensured that the core content and structure of the paper remain intact, avoiding extensive changes. The adjustments primarily involve rephrasing and emphasizing key points to gradually shift the paper’s focus from a method-oriented presentation to one centered on research questions, and better highlighting the significance of the proposed long-short architectures for the RL community. We believe that these modifications not only address your suggestions but also enhance the clarity and persuasiveness of the paper.
> > > >
> > > > >"I strongly encourage incorporating all the additional discussions and experiments from the rebuttal into the final revised manuscript."
> > > >
> > > > Thank you for all your valuable suggestions, which have provided deeper insights and enriched the broader research community. We have updated the PDF submitted according to all reviews from the rebuttal, including further clarification of the training process (Section 3.2, lines 237–238) and potential improvement for dimension ratio (Appendix F, lines 1306–1314), the empirical analysis of computational efficiency (Appendix E, lines 1269–1304) as well as other refinements and experiments throughout the manuscript to enhance clarity and completeness.  As we are unable to make further modifications to the PDF at this stage, we are happy to incorporate your valuable suggestions from this discussion into the camera-ready version.
> > > >
> > > >
> > > > We greatly appreciate your constructive feedback and consideration. We especially appreciate that the reviewers in ICLR not only raised concerns and questions but, more importantly, provided valuable suggestions to help us further strengthen our work.

---

> > > > > ### Comment · Reviewer_wr6s · 2024-12-03
> > > > > **Official Response by Reviewer wr6s to Authors**
> > > > >
> > > > > Dear Authors,
> > > > >
> > > > > I have thoroughly reviewed your latest response regarding how you plan to implement the suggested improvements. I appreciate the consideration you have given to reframing the manuscript to better emphasize the underlying research challenges, particularly regarding Markovian and non-Markovian properties in offline RL. The proposed additions to the Introduction, Motivation, and Conclusion sections better highlight the paper's key research questions and contributions while maintaining the presentation structure of the manuscript.
> > > > >
> > > > > Given that the OpenReview no longer allows revision updates at this stage, I would encourage you to consider providing an anonymous link to the revised manuscript for us reviewers and also ACs to reference.
> > > > >
> > > > > I look forward to seeing these improvements reflected in the camera-ready version and remain available for any further discussions that might benefit either the authors or my fellow reviewers.
> > > > >
> > > > > Best regards,
> > > > >
> > > > > Reviewer wr6s

---

> > > > > > ### Author Response · Authors · 2024-12-04
> > > > > >
> > > > > > Thank you once again for your time and thoughtful review of our response.
> > > > > > Following your suggestions, we have provided an anonymous link to the revised manuscript, which includes the additional discussions and experiments mentioned in our rebuttal. You can access the revised manuscript through the following link:
> > > > > >
> > > > > > https://github.com/anonymous-sub-paper/ICLR/blob/main/ICLR_2025_Conference_Submission.pdf
> > > > > >
> > > > > > To ensure clarity and facilitate your review, all improvements and further clarification have been highlighted in light blue. Please review the main text for changes and the appendix, where we have added new subsections with extended experiments and deep analyses.
> > > > > >
> > > > > > As the rebuttal phase comes to a close, we would like to sincerely thank all the reviewers for the constructive discussions and invaluable support during this period. Your feedback and engagement have been instrumental in improving our work and strengthening its contribution to the field. We truly appreciate the time and effort you have dedicated to this process! Thank you so much!

---

> ### Comment · Reviewer_vshZ · 2024-11-28
>
> Hi wr6s,
>
> Thank you for your insightful and constructive review.
> Your comments is exemplary, and have taught me a lot.
>
> Can I ask if you would like to update your rating, and why?
> I would be very grateful if you’re willing to share your insights.

---

> > ### Comment · Reviewer_wr6s · 2024-11-30
> > **Official Response by Reviewer wr6s to Reviewer vshZ**
> >
> > Dear fellow reviewer vshZ,
> >
> > Thank you for your kind words and interest in my review decision. I appreciate the collaborative spirit of your inquiry.
> >
> > I have decided to update my rating from 5 to 6 given the authors' positive and comprehensive rebuttal. This decision was based on several key factors:
> >
> > First, the authors provided clear and detailed responses addressing the technical concerns raised in my initial review. Their explanation of LSDT's unified training approach and the accompanying empirical evidence in Appendix B.3.2 resolved my questions about the convergence behavior of the dual-branch architecture.
> >
> > Second, while I initially had reservations about the dimension ratio adaptation, the authors presented a thoughtful analysis of the technical constraints and proposed alternative solutions through the HFM module. Their practical guidelines for selecting dimension ratios, supported by extensive experimentation, demonstrate a balanced approach between theoretical elegance and practical utility.
> >
> > Third, the authors' additional computational efficiency analysis provided valuable insights for practitioners, including comparisons of FLOPs, parameters, and memory usage across different network architectures. This practical implementation perspective was previously missing from the paper.
> >
> > As such, I believe these improvements collectively strengthen the paper's contribution to the field, justifying the updated rating. The work now appears to present a more complete and well-supported approach to combining global and local dependencies in RL decision transformers.
> >
> > Apart from these improvements, I have also suggested further enhancements for the final revised manuscript, particularly recommending a reframing to better emphasize the underlying research challenges, especially regarding Markovian and non-Markovian properties in RL. From my perspective, shifting from a method-oriented to a research question-oriented presentation would significantly enhance the paper's impact on the RL community, as the underlying question is ultimately more crucial than the method itself. I may further increase my rating according to the authors' response.
> >
> > I also look forward to further discussions and hearing your insights on this submission.
> >
> > Best regards,
> >
> > Reviewer wr6s

---

> ### Comment · Reviewer_vshZ · 2024-11-30
>
> Dear reviewer wr6s,
>
> I really appreciate your thoughtful comments and detailed feedback.
>
> As a first-time reviewer of ICLR, your comments has been a very important asset for me to learn from.
>
> I sincerely hope I can write insightful reviews as you one day.
>
> Thank you again for your response.
>
> All the best,
>
> vshZ

---

> > ### Comment · Reviewer_wr6s · 2024-11-30
> > **Official Response by Reviewer wr6s to Reviewer vshZ**
> >
> > Dear Reviewer vshZ,
> >
> > I appreciate your acknowledgment and positive attitude. Your generous comment about my review means a great deal to me, and I believe we all continue learning and improving as ICLR reviewers. Your comments about the routing mechanisms in the proposed LSDT has also raised important considerations that have enriched the overall evaluation.
> >
> > In my view, the peer review process works best when we engage in constructive dialogue and exchange of diverse perspectives. I look forward to further exchanges during the subsequent discussion phase as we work together to provide the authors and Area Chairs with constructive feedback possible for improving and fairly evaluating the submission.
> >
> > Best regards,
> >
> > Reviewer wr6s

---

### Meta-Review · Area_Chair_UEb7 · 2024-12-21

**Metareview:**

This paper presented a long-short decision transformer for generalizable decision-making. The proposed transformer architecture focuses on modeling both the local and global dependency for the task. A multi-head attention is used to model the global dependency and the dynamic convolution is utilized to model the local dependency. The structure is validated on different D4RL benchmarks and shows wide effectiveness. The three reviewers are generally positive about the submission. They acknowledged the interesting motivation, design, and the paper's experiments. The presentation of the paper is also clear. AC agrees with the comments from the reviewers and recommends accepting the paper as a poster.

**Additional Comments On Reviewer Discussion:**

The major comments mainly include some clarifications and additional ablation study experiments. During the rebuttal, the authors effectively addressed those comments, and the reviewers were generally satisfied with the authors' rebuttal.

---

### Decision · Program_Chairs · 2025-01-22

Accept (Poster)